# Identification of resistance mechanisms to small-molecule inhibition of TEAD-regulated transcription

Aishwarya Kulkarni [ID][1,2], Varshini Mohan [ID][1], Tracy T Tang [ID][3], Leonard Post[3], Yih-Chih Chan[1,2], Murray Manning[4,5], Niko Thio [ID][1], Benjamin L Parker[6], Mark A Dawson[1,2,7], Joseph Rosenbluh[4,5], Joseph HA Vissers [ID][1,2,7] & Kieran F Harvey [ID][1,2,8 ✉]

## Abstract

**The Hippo tumor suppressor pathway controls transcription by regulating nuclear abundance of YAP and TAZ, which activate transcription with the TEAD1-TEAD4 DNA-binding proteins. Recently, several small-molecule inhibitors of YAP and TEADs have been reported, with some entering clinical trials for different cancers with Hippo pathway deregulation, most notably, mesothelioma. Using genome-wide CRISPR/Cas9 screens we reveal that mutations in genes from the Hippo, MAPK, and JAK-STAT signaling pathways all modulate the response of mesothelioma cell lines to TEAD palmitoylation inhibitors. By exploring gene expression programs of mutant cells, we find that MAPK pathway hyperactivation confers resistance to TEAD inhibition by reinstating expression of a subset of YAP/TAZ target genes. Consistent with this, combined inhibition of TEAD and the MAPK kinase MEK, synergistically blocks proliferation of multiple mesothelioma and lung cancer cell lines and more potently reduces the growth of patient-derived lung cancer xenografts in vivo. Collectively, we reveal mechanisms by which cells can overcome small-molecule inhibition of TEAD palmitoylation and potential strategies to enhance the anti-tumor activity of emerging Hippo pathway targeted therapies.**

**Keywords** TEAD Inhibitors; Hippo Pathway; Cancer; Mesothelioma; Drug Resistance
**Subject Categories** Cancer; Chromatin, Transcription & Genomics; Signal Transduction

## Introduction

The Hippo pathway is an important regulator of organ growth and is also mutated in human cancers (Harvey et al, 2013). First discovered in *Drosophila*, it is highly conserved throughout evolution including in humans (Harvey and Hariharan, 2012; Zheng and Pan, 2019). The Hippo pathway has been predominantly studied in epithelial tissues, but also functions in other tissue types, including muscle, the nervous system, and some blood cell types (Halder and Johnson, 2011). Early evidence of a role for the Hippo pathway in cancer came from studies whereby the human orthologue of Salvador (SAV1), a founding member of the *Drosophila* Hippo pathway, were found to be mutated in renal cancer cell lines (Tapon et al, 2002). More recently, large-scale analyses of human cancer samples have revealed that multiple Hippo pathway genes are mutated in a broad range of cancer types (Wang et al, 2018). While Hippo pathway gene perturbations are relatively rare in most cancer types, they are very common in select cancers, such as mesothelioma, meningioma and squamous epithelial cell cancers (Harvey et al, 2013; Kulkarni et al, 2020; Wang et al, 2018).

The Hippo pathway responds to many cell biological properties such as cell-cell adhesion, cell-extracellular matrix adhesion, cell polarity and mechanical cues (Davis and Tapon, 2019). It can also respond to various stresses such as changes in osmolarity, temperature and energy stress (Koo and Guan, 2018). The pathway is comprised of more than 40 proteins, that relay information from the plasma membrane to the nucleus to modulate transcription. Typically, these signals are transduced by a core kinase cassette consisting of the MST1/2 kinases (Hippo in *Drosophila*) and LATS1/2 kinases (Warts in *Drosophila*). The activity of these kinases is controlled by scaffold proteins such as SAV1, Mob1A/B (Mats in *Drosophila*), and Merlin (encoded by the neurofibromatosis type 2 gene *NF2*). These proteins control transcription by regulating the rate at which the YAP and TAZ (Yorkie in *Drosophila*) transcription co-activator proteins move between the nucleus and cytoplasm (Manning et al, 2020). YAP and TAZ regulate transcription by binding to the TEAD1-TEAD4 DNA-binding proteins (Scalloped in *Drosophila*) and by recruiting transcriptional regulatory proteins such as the COMPASS complex, the Mediator complex and by promoting transcription elongation. TEAD1-4 can also repress transcription, together with the VGLL4 and INSM1 (*Drosophila* Tgi and Nerfin-1, respectively) transcription repressor proteins (Zheng and Pan, 2019). In addition,

[1]Peter MacCallum Cancer Centre, Melbourne, VIC 3000, Australia. [2]Sir Peter MacCallum Department of Oncology, The University of Melbourne, Parkville, VIC 3010, Australia. [3]Vivace Therapeutics Inc., San Mateo, CA 94404, USA. [4]Department of Biochemistry, and Biomedicine Discovery Institute, Monash University, Clayton 3800, Australia. [5]Functional Genomics Platform, Monash University, Clayton, VIC 3800, Australia. [6]Department of Anatomy & Physiology, The University of Melbourne, Parkville 3010 VIC, Australia. [7]Centre for Cancer Research and Department of Clinical Pathology, The University of Melbourne, Parkville, VIC 3010, Australia. [8]Department of Anatomy and Developmental Biology, and Biomedicine Discovery Institute, Monash University, Clayton 3800, Australia. ✉E-mail: kieran.harvey@petermac.org

*Drosophila* tissues, Yki increases Scalloped DNA dwell times on DNA, while Tgi and Nerfin-1 antagonize this (Manning et al, 2024).

Given the prevalence of Hippo pathway deregulation in many cancers, the dramatic tissue overgrowth that is caused by mutation of Hippo pathway genes in model organisms like *Drosophila* and mice, and the paucity of effective therapies for cancers like mesothelioma, the Hippo pathway is considered a potential therapeutic target. Interestingly, unbiased screening efforts by both pharma and academia have identified TEAD1-4 as druggable Hippo pathway proteins (Calses et al, 2019; Dey et al, 2020; Pobbati et al, 2023). Many compounds that influence TEAD-regulated transcription work by modulating auto-palmitoylation of a hydrophobic pocket in TEADs that normally stabilizes them and promotes their physical association with YAP and TAZ (Chan et al, 2016; Holden et al, 2020). The other major class of TEAD inhibitors (TEADi) that have been developed to date are those that bind to the surface of TEAD1-4 and directly impede their ability to bind to YAP (Chapeau et al, 2024). Multiple high potency TEADi have now been developed and subsequently entered clinical trials for cancers such as mesothelioma, as well as other cancers that harbor *NF2* mutations or oncogenic YAP gene fusions (Calses et al, 2019; Dey et al, 2020; Pobbati et al, 2023). Although targeted therapies have revolutionized cancer treatment, therapy resistance is inevitable in almost all patients and thus limits their long-term effectiveness (Garraway and Janne, 2012). As such, although Hippo-targeted therapies are predicted to be powerful anti-cancer agents, some degree of resistance to them is likely. To explore mechanisms of cell-intrinsic resistance to TEADi, and thus identify potential therapies that can be used in combination with TEADi, we performed whole genome CRISPR/Cas9 screens in mesothelioma cells. Multiple genes were identified that modulate the cellular response to TEAD inhibitors, most notably members of the Hippo, MAPK, and JAK/STAT pathways, thus identifying potential combination therapy approaches to be coupled with TEADi.

## Results

The discovery that TEAD transcription factors are post-translationally modified via auto-palmitoylation, and the importance of this for YAP/TAZ-TEAD driven transcription (Chan et al, 2016; Noland et al, 2016), has aided the development of small-molecule inhibitors of TEADs. The majority of these inhibitors block auto-palmitoylation of key cysteine residues in TEADs, impact the ability of TEADs to bind to YAP, and/or block proliferation of cancer cells with Hippo pathway mutations (Bum-Erdene et al, 2019; Chan et al, 2016; Gibault et al, 2021; Gridnev et al, 2022; Holden et al, 2020; Hu et al, 2022; Kaneda et al, 2020; Laraba et al, 2023; Li et al, 2020; Lu et al, 2021; Lu et al, 2023; Lu et al, 2019; Noland et al, 2016; Pobbati et al, 2015; Sun et al, 2022; Tang et al, 2021). Here, we used a range of approaches including large-scale functional genomics screens, and transcriptome profiling to investigate the mechanism of action of a potent small molecule inhibitor of all four TEADs (VT107), developed by Vivace Therapeutics (Tang et al, 2021). After confirming the nanomolar-potency of VT107 in the Hippo-pathway mutant mesothelioma cell lines NCI-H2052 (*NF2*, *LATS2* mutant) and NCI-H226 (*NF2*$^{-/-}$) (Miyanaga et al, 2015; Murakami et al, 2011; Sekido et al, 1995)

(Fig. 1A,B), we used immunoblotting, RNA sequencing (RNA-seq) and proteomics analyses to validate its on-target activity (Figs. 1C–H and EV1, Datasets EV1 and EV2). Gene expression was strongly impacted in both cell lines by VT107 and many known YAP/TAZ-TEAD target genes (e.g., *NPPB, IGFBP3, SNAPC1, CTGF, CYR61,* and *ANKRD1*) were among the top downregulated genes (Fig. 1D,E) (Cordenonsi et al, 2011; Hao et al, 2008; Varelas et al, 2008; Zanconato et al, 2015; Zhao et al, 2008). This was further confirmed by Gene set enrichment analysis (GSEA), which indicated that YAP activity was strongly reduced by VT107 in both cell lines (Figs. 1F and EV1C) (Cordenonsi et al, 2011; Subramanian et al, 2005). Finally, immunoblotting and proteomics experiments revealed that the expression of YAP/TAZ targets was reduced at the protein level by VT107 (Figs. 1C,G,H and EV1E, F, and Dataset EV2).

In addition to limiting the expression of YAP-TEAD target genes, VT107 modulated the expression of many genes that are not known to be regulated by YAP-TEAD. Most notably, genes that are responsive to MAPK pathway activity associated with KRAS hyperactivation, as well as genes induced by cytokine signaling (e.g., IL2 and IL15) were strongly upregulated upon VT107 treatment, while genes linked to the serum response, Hedgehog pathway and ribosome biogenesis were downregulated (Fig. 1F). To investigate whether acute inhibition of TEAD palmitoylation induces MAPK pathway genes by stimulating MAPK pathway signaling activity, we investigated the activity status of several key pathway proteins. We performed these experiments with VT107 and also with VT108, a chemical analog of VT107 that has similar biological activity (Haderk et al, 2024). The phosphorylation status and therefore activity of the MAPK pathway effector ERK was not increased in both NCI-H2052 and NCI-H226 cells upon 24-hour treatment with either TEAD palmitoylation inhibitor (VT107 or VT108) (Fig. 1I). Further, we observed no changes in the activity status of key signaling proteins from the Insulin and TOR pathways that can also be influenced by the MAPK pathway (Fig. 1I). Similarly, acute VT107 or VT108 treatment did not impact MAPK pathway activity in three additional mesothelioma cell lines (Fig. EV1G). Therefore, inhibition of TEAD palmitoylation must induce expression of MAPK pathway target genes via a mechanism that is independent of direct MAPK signaling pathway stimulation.

### Unbiased identification of genes that modulate the cellular response to TEAD inhibitors

To identify genes that either enhance or suppress the cellular response to VT107, we performed genome-wide CRISPR/Cas9 screens in both NCI-H2052 (H2052) and NCI-H226 (H226) cells (Fig. 2A). Each screen was conducted in duplicate using the Brunello CRISPR library and two doses of VT107 applied sequentially for up to two weeks per dose (Doench et al, 2016; Doench et al, 2014). Cells were treated first with an IC$_{50}$ dose (18 nM in H2052 cells, 32 nM in H226) and then with a cytostatic dose (100 nM for each cell line). Whole-genome sequencing was performed on genomic DNA extracted at the endpoint of vehicle (DMSO) or VT107 treatment (at both concentrations) and sgRNA abundance quantified. Using the MAGeCK algorithm we first compared sgRNA abundance in DMSO-treated cells (Li et al, 2014). The abundance of sgRNAs targeting known core cell essential genes (Hart et al, 2015) was decreased at the screening endpoint

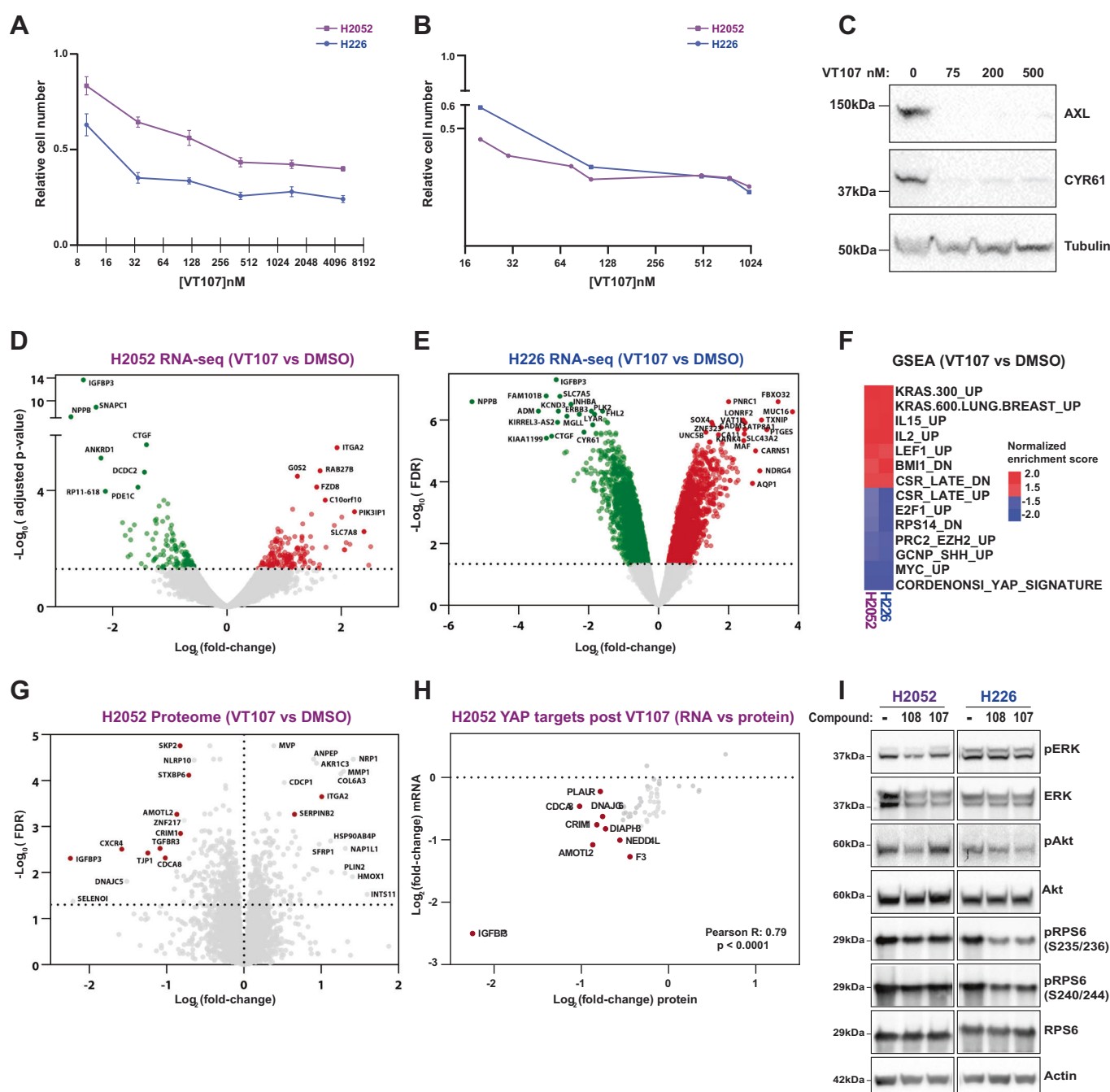

**Figure 1. The effect of TEAD inhibitors on the transcriptome and proteome of Hippo pathway mutant mesothelioma cells.**

(A, B) Charts showing H2052 and H226 cell numbers following treatment with different doses of VT107 for 6 days in: (A) 384-well plates ($n = 3$ biological replicates); or (B) T225 flasks, as used in CRISPR/Cas9 screening conditions ($n = 1$). (C) Immunoblots of lysates from H2052 cells treated with different doses of VT107 for 6 days and probed with the indicated antibodies. Molecular mass markers are indicated. (D, E) Volcano plots of gene expression of H2052 and H226 cells following 24-hour VT107 treatment, $n = 3$ biological replicates. Genes whose expression was strongly modulated are labeled. (F) A heatmap of strongly modulated transcription signatures by VT107 in H2052 and H226 cells, as determined by gene set enrichment analysis of differentially expressed genes (VT107 vs DMSO). (G) A volcano plot of protein expression of H2052 cells following 24-h VT107 treatment, $n = 5$ biological replicates. YAP/TAZ-TEAD targets are highlighted in red. (H) A correlation analysis of the effect of VT107 on mRNA and protein abundance of YAP/TAZ-TEAD targets. All plotted targets have significant protein level changes (q-value ≤ 0.05) in response to VT107. Correlation was assessed using the Pearson's correlation coefficient; R: 0.79, $p < 0.001$. The most significantly changed targets are labeled. (I) Immunoblots of lysates from the indicated cell lines, treated with 3 μM of VT107 or VT108 for 24 h and probed with the specified antibodies. Molecular mass markers are indicated. Data information: In (A), data are presented as mean ± SEM. In (D and E), significance was assessed using empirical Bayesian statistics. Source data are available online for this figure.

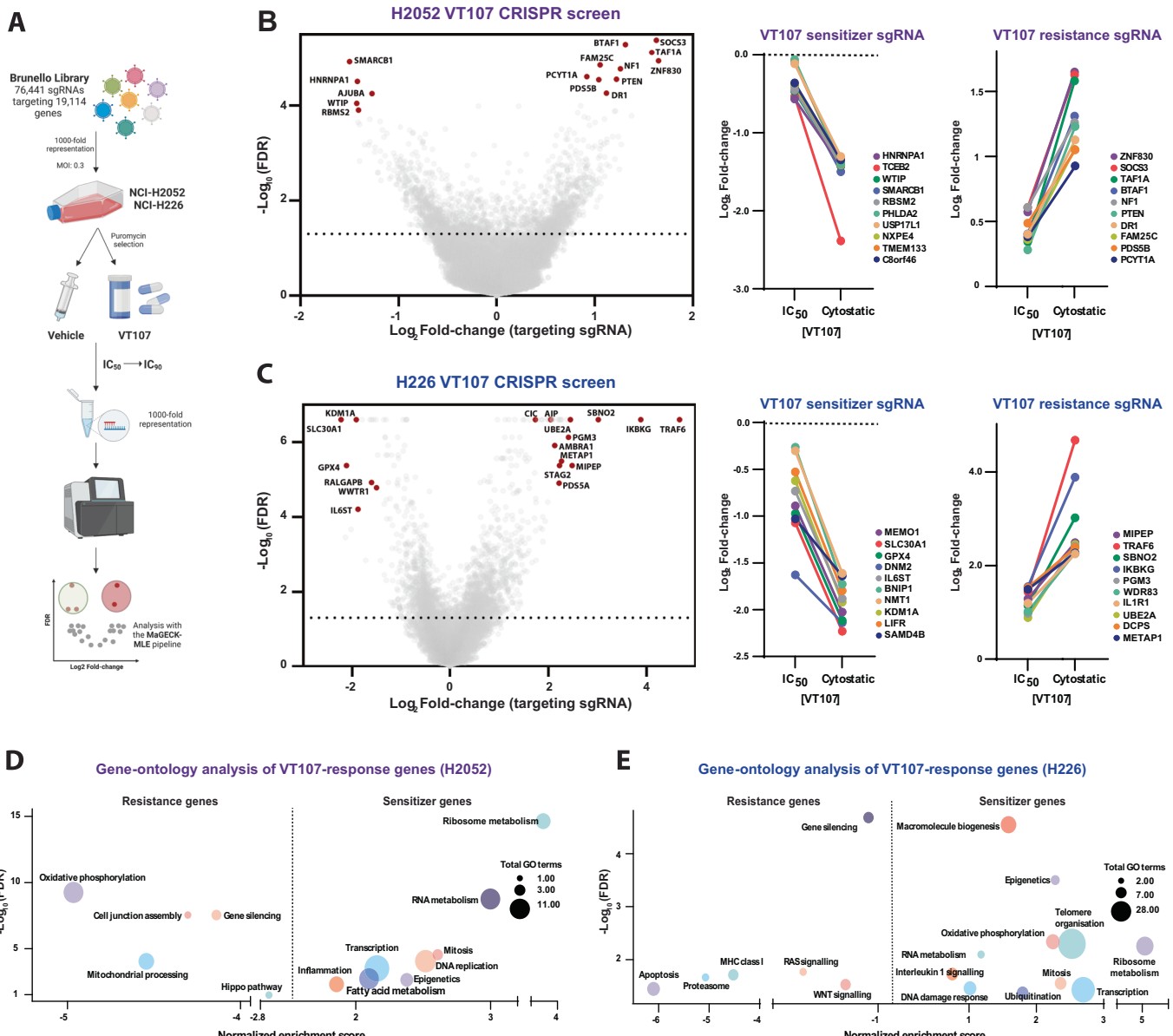

**Figure 2. Identification of genes that modulate the response of mesothelioma cells to TEAD inhibitors.**

(A) A schematic diagram outlining the protocol for the VT107 CRISPR/Cas9 screens. (B, C) Left panels: volcano plots representing the depletion or enrichment of targeting sgRNAs (per gene) following treatment with 100 nM VT107 (cytostatic dose) in CRISPR/Cas9 screens in H2052 cells (B) and H226 cells (C). Comparison of the depletion (middle panels) or enrichment (right panels) of targeting sgRNAs following treatment with VT107 at $IC_{50}$ and cytostatic doses in H2052 cells (B) or H226 cells (C) in CRISPR/Cas9 screens. (D, E) Bubble plots representing ontology analysis categories of genes that conferred VT107 resistance (targeting sgRNA $\log_2$Fold-change <0) or sensitivity (targeting sgRNA $\log_2$Fold-change > 0) in CRISPR/Cas9 screens in H2052 cells (D) or H226 cells (E). Each bubble is plotted according to the strongest scoring GO term from its respective category and total GO terms per category are indicated by bubble size. Data information: In (B and C), significance was assessed using empirical Bayesian statistics.

(Fig. EV2A,B and Dataset EV3), and we observed strong correlations in the gene log fold-changes in DMSO-treated populations with CRISPR/Cas9 screens of H2052 and H226 cells from the DepMap database, indicating the technical success of our screens (Fig. EV2C,D and Dataset EV4).

To identify genes that confer resistance to or with VT107 after CRISPR-mediated gene disruption, we used the MAGeCK algorithm to compare sgRNA abundance in DMSO-treated cells and VT107-treated cells. We filtered genes whose targeting-sgRNA showed significant (FDR ≤ 0.05) changes in abundance at both screening endpoints and applied a magnitude threshold to identify genes whose loss either conferred resistance to VT107 (targeting-sgRNA log fold-change ≥0.5 in VT107-treatment population relative to vehicle population) or sensitivity to VT107 (targeting-sgRNA log fold-change ≤ −0.5). This revealed 674 or 818 putative drivers of VT107 resistance and 571 or 519 VT107 sensitivity genes

in H2052 and H226 cells, respectively (Fig. 2B,C and Dataset EV5). Guide RNAs targeting the majority of VT107 sensitivity genes showed a positive fold-change at both the $IC_{50}$ and $IC_{90}$ timepoints, and similarly, gRNAs targeting most VT107 resistance genes exhibited negative log fold-changes at both doses (middle and left panels of Fig. 2B,C, and EV2E–H). This indicated that the response to VT107 was maintained for the duration of both screens. Putative sensitivity genes from both screens were enriched in the cellular processes of ribosome metabolism, RNA metabolism, and transcription, while some ontology groups were only enriched in one cell line (Fig. 2D,E).

By comparing the H2052 and H226 CRISPR/Cas9 screens, we found a weak positive correlation (Pearson's R: 0.1921, $p < 0.01$), further indicating shared mechanisms in how these cells respond to VT107, but also differences (Fig. 3A). Common sensitivity and resistance genes from each screen were identified, as well as common gene ontology groups (Fig. 3A–C). Among these, several key Hippo pathway genes modulated the response to VT107, most notably *VGLL4* (whose loss promoted VT107 resistance) and *WWTR1 (*the loss of which promoted VT107 sensitivity) (Fig. 3A,B). *VGLL4* encodes a transcriptional repressor protein that competes with YAP for TEAD1-4 binding, while *WWTR1 (*also known as *TAZ)* is a paralog of *YAP* (Hong et al, 2005; Koontz et al, 2013). Upstream Hippo pathway genes that promote pathway activity were not identified as conferring resistance to VT107, possibly because the Hippo pathway is already strongly perturbed in H2052 and H226 cells, owing to *NF2* and *LATS2* mutations and/or because of functional redundancy between homologous genes (Miyanaga et al, 2015; Murakami et al, 2011; Sekido et al, 1995). In fact, the loss of some upstream Hippo pathway genes (e.g., *PTPN14*) counterintuitively conferred some degree of sensitivity to VT107 (Fig. 3B), the significance of which is currently unclear (Wang et al, 2012). In addition, sgRNA-based depletion of the LATS kinase repressors WTIP, AJUBA, and TRIP6 conferred sensitivity to VT107 in H2052 but not H226 cells (Fig. 2B and Dataset EV5). Strikingly, two major negative regulators of the MAPK pathway and bona fide human tumor suppressor genes, *Neurofibromin 1 (NF1)* and *Capicua (CIC)*, conferred strong resistance to VT107 when depleted by sgRNA in both cell lines (Fig. 3A,B) (Cichowski and Jacks, 2001; Kawamura-Saito et al, 2006; Kim et al, 2021; LeBlanc et al, 2017; Ratner and Miller, 2015; Xu et al, 1990). In addition, inactivation of the JAK-STAT pathway effector *STAT3* conferred strong sensitivity to VT107, while that of the JAK-STAT pathway repressor *SOCS3* elicited strong resistance in H2052 cells (Fig. 3A,B) (Rawlings et al, 2004). Collectively, genome-wide CRISPR/Cas9 screens indicated that modulation of multiple genes and signaling pathways can modify the response of mesothelioma cells to small molecule inhibition of YAP/TEAD-mediated transcription.

## Inactivation of MAPK, JAK/STAT, and Hippo pathway repressors confers resistance to TEAD inhibitors

To validate our whole-genome screens, we used CRISPR/Cas9 mutagenesis with two independent sgRNAs to mutate *NF1*, *SOCS3,* and *VGLL4* in H226 and H2052 cells, as well as a third mesothelioma cell line, MSTO-211H, which harbors inactivating mutations in both *LATS1* and *LATS2* (Miyanaga et al, 2015). A substantial reduction of NF1 and VGLL4 protein was confirmed by western blot in all 6 mutant cell lines (Fig. 3D). Endogenous SOCS3

protein could not be detected by western blot so we used Tracking of Indels by Decomposition (TIDE) analysis to confirm CRISPR/Cas9 mutagenesis at the *SOCS3* locus in all sg*SOCS3*-transduced cell lines (Fig. EV3A–F). Additionally, we assessed JAK/STAT pathway activation using STAT3 and p-STAT3 antibodies (STAT3 becomes phosphorylated on Y705 when the pathway is activated) (Heinrich et al, 1998) and observed strong JAK/STAT pathway activation in *SOCS3* mutant H2052 and MSTO-211H cells, while pathway activity was already elevated in H226 cells (Fig. 3E). *NF1* loss caused strong MAPK pathway hyperactivation in both H2052 and H226 cells (as assessed by phosphorylation of MEK and ERK at S221 or T202/Y204, respectively), while MSTO-211H cells exhibited high baseline MAPK pathway activity (Fig. 3F). Importantly, mutation of *NF1* or *SOCS3* with both sgRNAs induced VT107 resistance in all three cell lines, while mutation of *VGLL4* with one sgRNA conferred resistance to VT107 in all three cell lines, and in 2 out of 3 cell lines with the other sgRNA (Fig. 3G). To assess whether these genetic mutations impacted the response of mesothelioma cells to other TEAD palmitoylation inhibitors, we performed similar experiments with VT104 (an independent TEAD palmitoylation inhibitor developed by Vivace Therapeutics) and VT106 (a less active enantiomer of VT107) (Tang et al, 2021). Loss of *NF1*, *VGLL4*, and *SOCS3* in NCI-H2052 cells all conferred similar degrees of resistance to VT104, as they did to VT107, and as expected, the viability of none of these cell lines was impacted by VT106 (Fig. EV4A). This indicates that the response of *NF1*, *VGLL4*, and *SOCS3* mutant NCI-H2052 cells to inhibition of TEAD palmitoylation is not specific to VT107.

## Mutation of *NF1* reinstates expression of select YAP/TAZ-TEAD target genes

Next, we sought to investigate how resistance genes identified in our CRISPR/Cas9 screens modulate the cellular response to TEAD inhibition. Given the close functional links between the MAPK and Hippo pathways in transcription and cancer (Herranz et al, 2012; Koo et al, 2020; Lin et al, 2015; Park et al, 2020; Pascual et al, 2017; Pham et al, 2021; Reddy and Irvine, 2013; Stein et al, 2015; Zanconato et al, 2015), we investigated this in *NF1* mutant H2052 cells, which displayed MAPK pathway activation (Fig. 3F). Parental and *NF1* mutant H2052 cells were treated with DMSO or VT107, and RNA harvested and sequenced 24 h later. As expected, the transcriptome of *NF1* mutant cells differed significantly from parental cells in the baseline (DMSO-treated) condition (Figs. 4A and EV4B,C). Unbiased gene set enrichment analyses revealed elevation of transcriptional signatures linked to the WNT and Estrogen response pathways in *NF1* mutant cells, consistent with published reports (Luscan et al, 2014; Zheng et al, 2020). Transcription signatures associated with oxidative phosphorylation and fatty acid metabolism were also elevated in *NF1* mutant cells (Fig. 4B). Next, we compared the transcriptional response to VT107 in parental and *NF1* mutant H2052 cells. VT107 had a more profound effect on the transcriptome of parental cells compared to *NF1* mutant cells; expression of 564 genes was significantly altered by 24 h of VT107 treatment in parental cells while only 163 genes were altered by VT107 in *NF1* mutant cells (Figs. 1D and 4C,D and Dataset EV6).

Given that the Hippo and MAPK pathways co-regulate many genes (Koo et al, 2020; Liu et al, 2016; Obier et al, 2016; Park et al, 2020; Pascual et al, 2017; Pham et al, 2021; Stein et al, 2015;

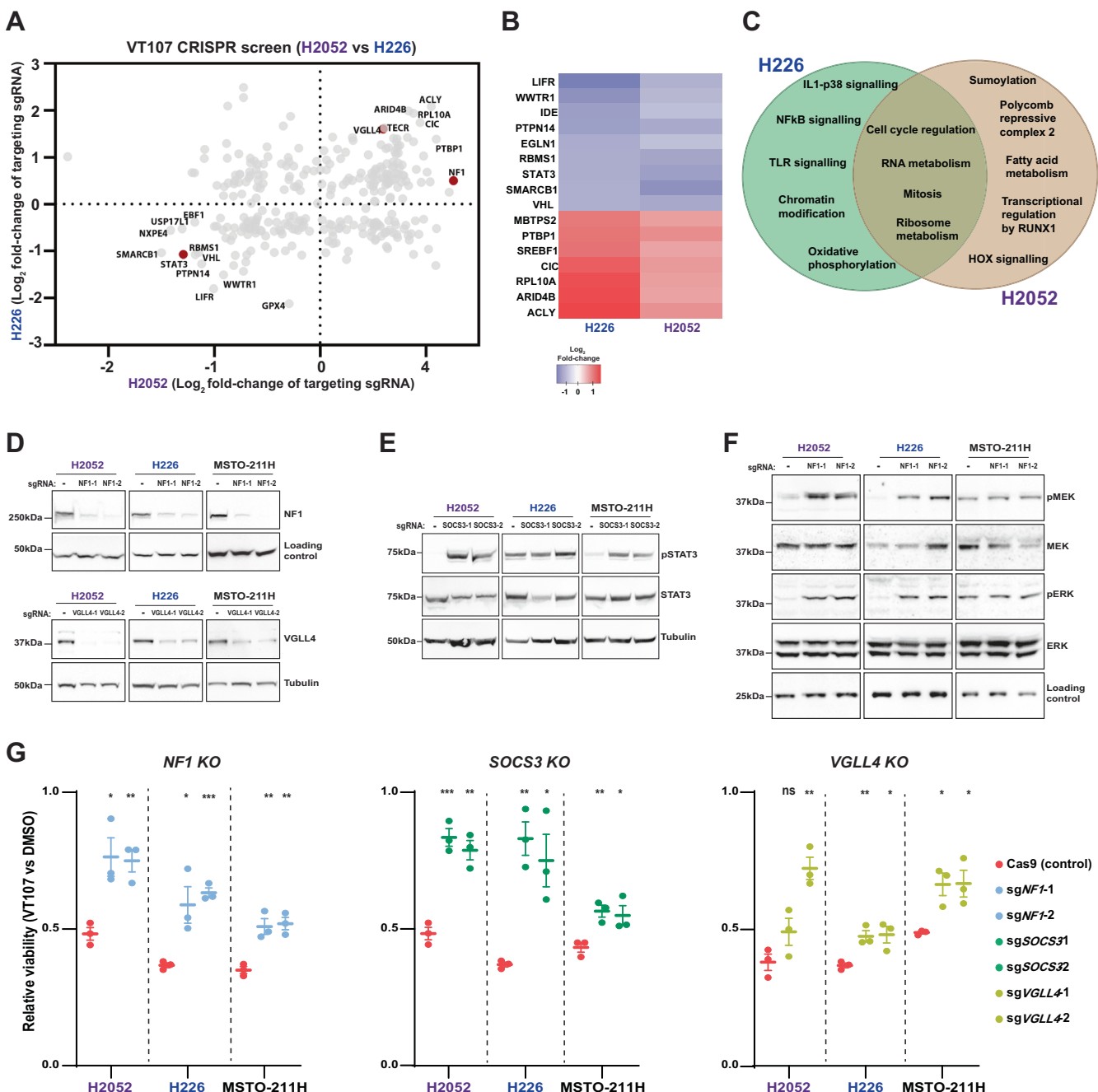

**Figure 3. Inactivation of MAPK, JAK/STAT, and Hippo pathway repressors confer resistance to TEAD inhibitors.**

(A) A correlation plot comparing the response to VT107 treatment in CRISPR/Cas9 screens in H2052 and H226 cells. Genes that are highlighted in red were studied subsequently. Correlation was assessed using the Pearson's correlation coefficient. (B) A heatmap of high stringency VT107 resistance genes (targeting sgRNA log₂Fold-change ≤ −0.8) and sensitivity genes (targeting sgRNA log₂Fold-change ≥ 0.8) identified in CRISPR/Cas9 screens in both H2052 and H226 cells. (C) A Venn diagram of GO terms that were enriched among VT107 resistance or sensitivity genes in CRISPR/Cas9 screens in either H2052 or H226 cells alone, or common to both screens. (D–F) Immunoblots of lysates from H2052, H226, and MSTO-211H cells expressing Cas9. Cells were control parental cells or expressed the following independent gRNAs: *NF1* (D, top panel and F); *VGLL4* in (D, bottom panel); *SOCS3* (E). Lysates were probed with the indicated antibodies. Equal protein loading was confirmed by detection of Tubulin or by non-specific bands from the anti-NF1 antibody in the top panel (D), or the anti-pERK antibody (F). Molecular mass markers are indicated. (G) Charts of the impact of VT107 on the viability of parental and mutant H2052, H226, and MSTO-211H cells, as assessed by alamar blue assays. H2052 and H226 cells were treated with 100 nM VT107 and MSTO-211H cells with 370 nM VT107 for 4 days. Genes that were mutated were *NF1* (left panel), *SOCS3* (middle panel), or *VGLL4* (right panel). $n = 3$ biological replicates. Data information: In (G), data are presented as mean ± SEM. *$p < 0.05$, **$p < 0.01$, ***$p < 0.001$, ns—not significant (Student's t-tests). P values were (from left to right): NF1—0.0192; 0.0046; 0.0310; 0.0002; 0.0072; 0.0028; SOC3—0.0009; 0.0020; 0.0017; 0.0168; 0.0090; 0.0425; VGLL4—0.1249; 0.0025; 0.0086; 0.0233; 0.0133; 0.0220. Source data are available online for this figure.

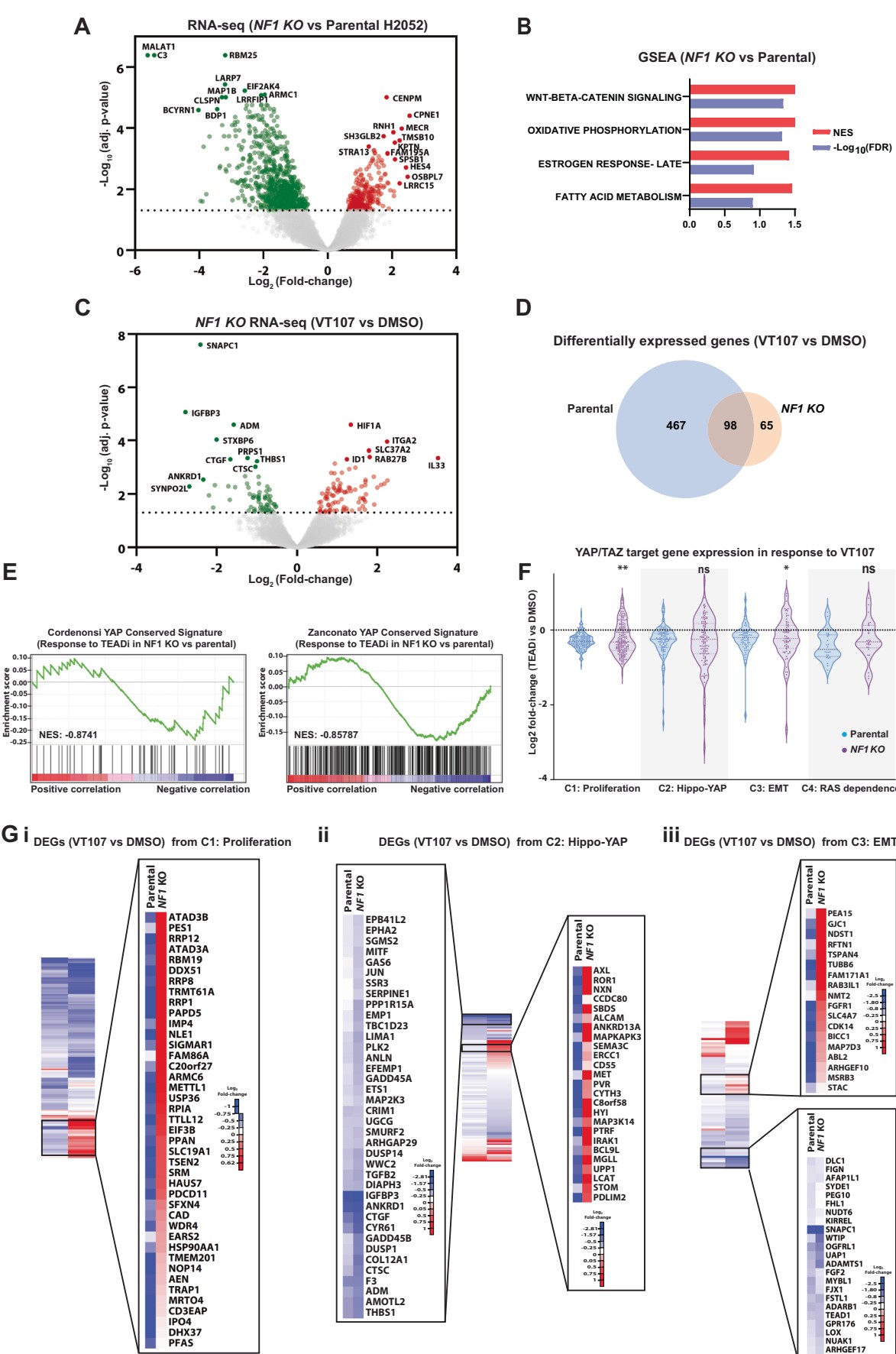

**Figure 4. Mutation of *NF1* in mesothelioma cells reinstates expression of select YAP/TAZ-TEAD target genes.**

(A) A volcano plot of gene expression of *NF1* mutant cells relative to parental H2052 cells, n = 3 biological replicates. (B) A bar chart indicating top scoring transcription signatures from gene set enrichment analysis of differentially expressed genes in *NF1* mutant cells compared to parental cells. (C) A volcano plot of gene expression of *NF1* mutant H2052 cells (treated with DMSO or 1 μM VT107 for 24 h), n = 3 biological replicates. (D) A Venn diagram indicating the degree of overlap of differentially expressed genes in parental and *NF1* mutant cells treated with VT107. (E) Gene set enrichment analysis plots of two YAP transcription signatures among differentially expressed genes in VT107-treated *NF1* mutant compared to parental H2052 cells. (F) Violin plots of differentially expressed genes (VT107 vs DMSO) in both parental and *NF1* mutant H2052 cells. Four independent clusters of genes that are sensitive to YAP/TAZ expression were assessed, n = 3 biological replicates. (G) Heatmaps of relative gene expression level changes (VT107 vs DMSO) in parental and *NF1* mutant H2052 cells. Heatmaps are shown for genes from cluster 1 (i), cluster 2 (ii) and cluster 3 (iii). Magnified regions focus on select genes from clusters 1–3 whose expression was restored in VT107-treated *NF1* mutant, compared to parental, H2052 cells. Also magnified are select genes from clusters 2 and 3 whose expression was not restored by *NF1* loss; these include many core YAP/TAZ-TEAD target and Hippo pathway genes. Data information: In (F), the null hypothesis for differences between parental and *NF1* mutant cells was tested with a hypergeometric distribution analysis **p < 0.01, *p < 0.05, ns—not significant. *P* values were (from left to right): 0.0005; 0.1247; 0.044; 0.3559. In (A, C), significance was assessed using empirical Bayesian statistics.

Zanconato et al, 2015), and VT107's impact on the transcriptome was blunted in *NF1* mutant H2052 cells, we considered the possibility that expression of YAP-TEAD target genes was restored in VT107-treated *NF1* mutant cells. To investigate this, we performed targeted GSEA with published YAP-TEAD target gene signatures (Cordenonsi et al, 2011; Pham et al, 2021; Zanconato et al, 2015). Subsets of genes from two independent YAP-TEAD signatures were depleted less in VT107-treated *NF1* mutant cells relative to parental cells, implying weakly restored expression of YAP-TEAD target genes (Fig. 4E). To examine this further, we assessed the expression of four published clusters of genes that were recently reported to be strongly associated with depletion of YAP/TAZ across multiple cell types (Pham et al, 2021). Functionally, these clusters are most closely related to: (1) cell proliferation, including genes regulated by the Myc oncoprotein; (2) Hippo signaling (Hippo-YAP); (3) epithelial to mesenchymal transition (EMT); and 4) MAPK signaling (RAS dependence) (Pham et al, 2021). By comparing the magnitude of expression changes of each gene in all four clusters upon VT107 treatment, we found that select genes from each cluster were restored in *NF1* mutant cells (Fig. 4F). Statistical analyses revealed that cluster 1 genes and cluster 3 genes, when considered as populations, were significantly restored in *NF1* mutant cells treated with VT107, compared to parental cells (Fig. 4F). Finally, we examined expression changes in individual genes from select clusters, which underscored the observation that only select genes that are responsive to YAP/TAZ-regulated transcription were restored in *NF1* mutant cells upon VT107 treatment (Fig. 4G). Interestingly, the expression of many genes from cluster 2 that are considered "core" YAP/TAZ-TEAD target genes and are often used as surrogates for YAP activity (e.g., *CTGF, CYR61, ANKRD1, AMOTL2*) were not restored in *NF1* mutant cells, while many other genes were (Fig. 4G). Similarly, many genes that encode Hippo pathway proteins from the EMT cluster (e.g., *TEAD1, WTIP, FJX1, ARHGEF17*, and *KIRREL*) were not restored in VT107-treated *NF1* mutant H2052 cells. Collectively, these studies suggest that VT107 resistance in *NF1* mutant cells was driven by partial reinstatement of genes that are sensitive to YAP/TEAD activity.

## Combined MAPK and Hippo pathway inhibition synergistically impacts mesothelioma cells

Next, we sought to determine whether combining MAPK pathway or JAK/STAT pathway inhibition with VT107 enhanced its ability to inhibit mesothelioma cell proliferation. For this, we measured the impact of combining VT107 treatment with either the MEK1/2 inhibitor trametinib or 5 independent JAK-STAT inhibitors that target different components of the IL6/Gp130-JAK-STAT3 pathway (Fig. 5A). In both H2052 and H226 cell lines, VT107 and trametinib synergistically inhibited cell proliferation (Fig. 5B–D), which is consistent with several published studies on genetic or chemical targeting of the Hippo and MAPK pathways (Kim et al, 2016; Koo et al, 2020; Lin et al, 2015; Liu et al, 2016; Park et al, 2020; Zanconato et al, 2015). In contrast, while the JAK1/2 inhibitor AZD1480 exhibited synergistic or strong additive interactions with VT107 in H226 and H2052 cells (Fig. 5B,E,F), the other JAK-STAT pathway inhibitors we tested exhibited weak additive interactions when combined with VT107 in both cell lines (Figs. 5B and EV5). Next, we investigated how general the impact of combining Hippo/MAPK therapies is by expanding our studies to include additional mesothelioma cell lines and the independent but related TEADi, VT108. VT108 and trametinib synergistically impeded proliferation of four out of 10 mesothelioma cell lines tested and had additive impacts on the proliferation of the remaining 6 cell lines (Figs. 5G and EV6A).

## Combined MAPK and Hippo pathway inhibition synergistically impacts non-small cell lung cancer cells and tumors

To explore combined inhibition of the Hippo and MAPK pathways in non-mesothelioma cancers, we performed similar combination therapy studies on non-small cell lung cancer (NSCLC) cell lines. In addition, to determine whether the synergistic impact of combined Hippo/MAPK therapy was not limited to the MEKi trametinib, we utilized an independent MEKi, cobimetinib (Rice et al, 2012). Synergistic impacts between VT108 and cobimetinib were observed in ten out of 42 NSCLC cell lines tested, while additive effects were observed on the proliferation of the remaining 32 cell lines (Figs. 5H and EV6B). While TEADi and MEKi displayed synergy in both mesothelioma and NSCLC cell lines, synergism was observed at lower TEADi doses in mesothelioma cell lines (Fig. EV6C).

Given the strong support for combined Hippo and MAPK pathway inhibition as a cancer treatment in our studies and in the literature, we explored the effect of combined TEADi and MEKi treatment on cancer cell line-derived tumors grown in vivo. For this, we selected two patient-derived xenograft (PDX) models of *NF2* mutant NSCLC cells, as we observed TEADi-MEKi synergism in both NSCLC and mesothelioma cell lines in vitro (Figs. 5, EV5 and EV6), and Hippo mutant mesothelioma cells either did

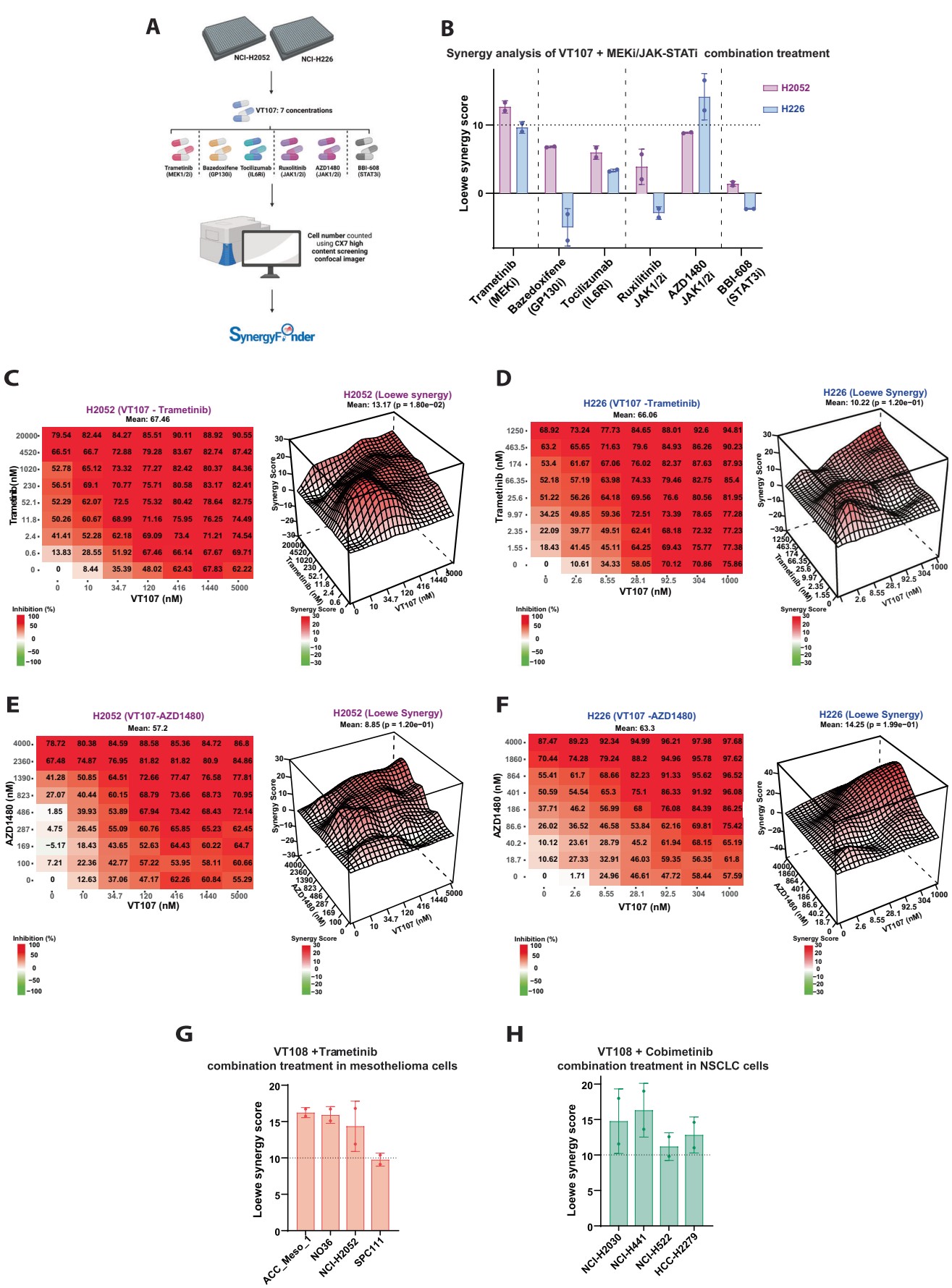

◄ **Figure 5. Combined MAPK and Hippo pathway inhibition synergistically impacts mesothelioma and NSCLC cells.**

(A) A schematic diagram outlining the protocol used for combination drug treatment assays conducted in H2052 or H226 cells. (B) A bar chart indicating the interaction outcome of combined treatment with VT107 and the indicated compounds on H2052 and H226 cell numbers. Synergistic interactions were indicated by synergy scores ≥10 and additive interactions by scores within the range of −10 to 10. $n = 2$ biological replicates (2 technical replicates per biological replicate). (C–F) Left panels: dose-response matrices indicating the level of inhibition of cell number by treatment of cells with the indicated doses of VT107 and/or trametinib. Right panels: representative 3D synergy calculation topological maps indicating the type and degree of interaction between drugs on inhibition of cell number. H2052 cells were treated in (C) and (E), H226 cells were treated in (D) and (F). VT107 was combined with MEKi in (C) and (D), and with AZD1480 in (E) and (F). (G, H) Bar charts indicating the interaction outcome of combination treatment with VT108 and trametinib in mesothelioma cell lines in (G) or VT108 and cobimetinib in NSCLC cell lines in (H). In all assays, $n = 2$ technical replicates and error bars: SD. Data information: In (B, G, H), data are presented as mean ± SEM. In (C–F) mean synergy scores were tested for statistical significance using the parametric bootstrapping method, $n = 2$.

not form tumors in mice or were unsuitable for in vivo combination therapy experiments as they were extremely sensitive to TEADi monotherapy (Tang et al, 2021). The PDX lines we used were: (1) LU-01-0407 PDX, which has reduced *NF2* copy number variation; and (2) LU-01-0236 PDX, which has a homozygous E265 to stop codon mutation in *NF2*. VT108 was used for these studies as it has a longer half-life in mice than VT107. PDX tumor-bearing mice were treated with either vehicle, trametinib alone, VT108 alone, or a combination of both compounds, with doses defined based on preliminary experiments with each tumor type. As monotherapies, trametinib and VT108 reduced tumor growth to some extent in both PDX tumors (Fig. 6A,C). For LU-01-0407 PDX tumors, combination trametinib and VT108 therapy reduced tumor size significantly more than either agent alone and almost completely impeded tumor growth (Fig. 6A). For LU-01-0236 PDX tumors, combination trametinib and VT108 therapy reduced tumor growth significantly more than VT108 alone (Fig. 6C). In addition, we observed minimal unfavorable impact on the body weight of mice throughout the course of these experiments (Fig. 6B,D). Collectively, these tumor xenograft studies provide in vivo evidence for the benefit of treating Hippo mutant tumors with combination therapies that target both MEK and TEAD.

## Discussion

Following its discovery in *Drosophila* as a regulator of tissue growth, the Hippo pathway has been touted as a potential target for new cancer therapies (Dey et al, 2020). Two decades later, the first raft of Hippo pathway targeted therapies, which all inhibit YAP/TAZ-TEAD regulated transcription, have entered clinical trials (Calses et al, 2019; Pobbati et al, 2023). Here, to identify potential resistance mechanisms and combination approaches for Hippo-targeted therapies, we performed unbiased screens to investigate the cellular response to one such agent, the TEAD palmitoylation inhibitor VT107 (Tang et al, 2021). Whole genome CRISR/Cas9 screens of two mesothelioma cell lines that harbor Hippo pathway mutations identified multiple genes that confer either resistance or sensitivity to inhibition of TEAD palmitoylation. Among these were groups of genes that were unique to each cell line and belonged to ontology groups such as fatty acid metabolism (H2052 cells) and interleukin signaling (H226 cells). In addition, select gene ontology groups were enriched in both cell lines, including ribosome metabolism and RNA metabolism. Further exploration of these cellular processes is required to understand the molecular mechanisms by which they confer resistance or sensitivity to VT107. On an individual gene basis, a core set of

genes modified TEAD's anti-proliferative effect in both mesothelioma cell lines. Notable among genes that conferred resistance to VT107 was *VGLL4*, which encodes a transcription co-repressor that competes with YAP/TAZ for binding to TEADs. *VGLL4* loss conferred strong resistance to VT107, suggesting that VT107 induces transcription repression by VGLL4/TEAD and associated transcription co-repressors and this is important for VT107's ability to limit mesothelioma cell proliferation and viability.

Two additional cancer signaling pathways scored strongly in both CRISPR/Cas9 screens: the MAPK and JAK/STAT pathways. The loss of two bona fide MAPK pathway tumor suppressors conferred resistance to VT107: *NF1*, which functions in the upstream part of the signal transduction pathway; and *CIC*, a transcription repressor. From the JAK/STAT pathway the upstream signaling repressor SOCS3 conferred VT107 sensitivity, while the STAT3 transcription factor conferred resistance to VT107. Interestingly, both STAT3, and, in particular, the MAPK pathway-regulated AP-1 transcription factors, co-regulate transcription of many genes (He et al, 2021; Koo et al, 2020; Liu et al, 2016; Obier et al, 2016; Park et al, 2020; Pascual et al, 2017; Pham et al, 2021; Stein et al, 2015; Zanconato et al, 2015), suggesting a possible mechanism by why which genetic modulation of these pathways modify the cellular impact of VT107. Transcriptome profiling of *NF1* mutant H2052 mesothelioma cells indicated that VT107 strongly represses the expression of many YAP/TEAD target genes in both parental and *NF1* mutant cells, but the transcriptional response is blunted by *NF1* loss. On further inspection, subsets of genes that are sensitive to YAP/TAZ activity were partially restored by *NF1* mutation, suggesting that MAPK pathway hyperactivation confers resistance to VT107 by reinstating a subset of YAP/TEAD target genes. The mechanistic basis for this is currently unclear but could potentially be related to the fact that AP-1 transcription factors and YAP/TEAD co-regulate an overlapping transcriptome that is important in cell proliferation and tissue growth (Koo et al, 2020; Liu et al, 2016; Obier et al, 2016; Park et al, 2020; Pascual et al, 2017; Pham et al, 2021; Stein et al, 2015; Zanconato et al, 2015).

The findings of our CRISPR/Cas9 screens and subsequent mechanistic studies spurred us to investigate combination therapies between TEAD and drugs that target either the MAPK or JAK/STAT pathways. Indeed, combining different MEKi's and TEADi's synergistically or additively repressed the growth of multiple mesothelioma and NSCLC cell lines and PDX tumors grown in mice, while the JAKi AZD1480 and the TEADi VT107 showed synergistic or strong additive effects on mesothelioma cell lines. Thus, our experiments provide additional evidence that combining Hippo and MAPK pathway targeted therapies offers promise for

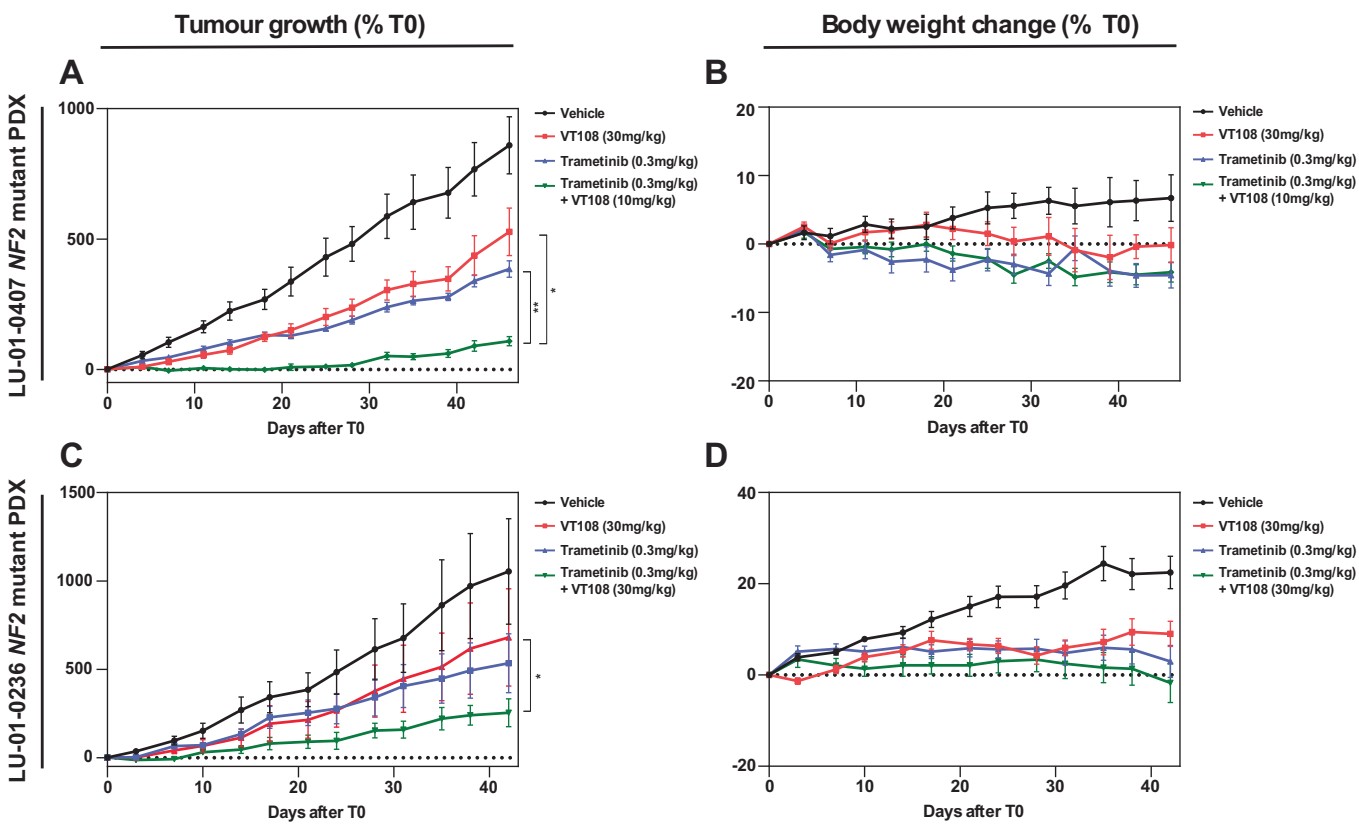

**Figure 6. Combined MAPK and Hippo pathway inhibition synergistically impacts NSCLC tumors.**

(A–D) The impact of VT108, cobimetinib, or both in combination on tumor growth in the NSCLC PDX models LU-01-0407 or LU-01-0236 (A, C), and the average body weight of mice from each treatment group (B, D). $n = 8$ for LU-01-0407, $n = 6$ for LU-01-0236. Data information: In (A–D), data are presented as mean ± SEM. In (A) *$p = 0.0149$, **$p = 0.001$, and in (B) *$p = 0.018$, one-way ANOVA (F-test) was performed on data from T42. Source data are available online for this figure.

the treatment of mesothelioma and other cancers such as NSCLC that have mutations in one or both pathways. We also identify the potential of combined targeting of the Hippo and JAK/STAT pathways in mesothelioma, although this requires additional validation. Further, our study identifies potential resistance mechanisms that might limit the effectiveness of TEAD palmitoylation inhibitors if they fulfill their promise and eventuate as therapies for one or more cancer types.

## Methods

### Cell culture

NCI-H2052, NCI-H226, and MSTO-211H cells (CVCL_1518, CVCL_1544, and CVCL_1430, respectively) were gifted by A/Prof Tom John (Olivia Newton John Cancer Centre, Melbourne, Australia). Cells were cultured in Roswell Park Memorial Institute Medium (Thermo Fisher Scientific) supplemented with 10% Fetal Bovine Serum (Hyclone, SH30396.03) and their mycoplasma-free status was confirmed every 3–6 months with standard PCR (5'YCGCTGVGTAGTATRYWCGC3', 5'GCGGTGTGTA-CAARMCCCGA3') (Uphoff and Drexler, 2011).

### Proliferation and viability assays

In proliferation assays, cells were seeded in T175 flasks (Cellstar) then treated with DMSO or different doses of VT107, VT104 or VT106 (Vivace Therapeutics). At 6 days of treatment, cell number was counted using a Coulter counter (Beckmann Coulter). In viability assays, cells were seeded in 96-well plates then treated with DMSO or different doses of VT107. Cell viability was measured 96-h after treatment by Alamar Blue assay. Measurements were taken at 540/610 nm with a Cytation 3 plate reader (Biotek).

### Immunoblotting

Immunoblotting was conducted as in (Zhang et al, 2020) and membranes probed with primary antibodies targeting AXL (RRID:AB_11217435), CYR61 (RRID:AB_2798492), Tubulin (RRID:AB_), p-ERK T202/204 (RRID:AB_331768 or RRID:AB_2315112), ERK (RRID:AB_330744), p-MEK S217/221 (RRID:AB_490903 or RRID:AB_2138017), MEK (RRID:AB_10695868), p-STAT3 (RRID:AB_2491009), STAT3 (RRI-D:AB_2798995), NF1 (RRID:AB_2798543), VGLL4 (RRID:AB_106774549), p-Akt S473 (RRID:AB_2315049), Akt (RRID:AB_915783), pRPS6 S240/244 (RRID:AB10694233), pRPS6 S235/236 (RRI-D:AB331679), pRPS6 (RRID:AB_331355), Actin (RRID:AB_2242334).

## RNA sequencing

Cells were treated with 1 µM VT107 or the equivalent %(v/v) DMSO and incubated for 24 h. Total RNA was harvested using Trizol reagent (Thermo Fisher Scientific) and quantified with the Qubit RNA HS (Thermo Fisher Scientific). Indexed libraries were pooled and sequenced on a NextSeq500 (Illumina) and 5–15 million single-end, 75 bp reads were generated per sample.

## Analysis of RNA-sequencing data

Differential gene expression (VT107 vs DMSO) was quantified per cell line using the voomwithQualityWeights or EdgeR pipelines on filtered sequencing data (reads with CPM ≥ 0.5 in all samples) (Liu et al, 2015; Ritchie et al, 2015). Where comparisons of gene-expression (VT107 vs DMSO) between cell lines were made, differential expression per cell line was quantified using the edgeR pipeline on filtered sequencing data (reads ≥10 in at least two biological replicates per group) or an interaction matrix using the limma-voom pipeline was applied. Ranked-list gene set enrichment analysis was conducted with standard protocol (Subramanian et al, 2005) using the Hallmark and Oncogenic gene set collections from the Molecular signatures database (MsigDB) (Liberzon et al, 2011) or independently selected YAP-TEAD target gene signatures from the literature (Cordenonsi et al, 2011; Zanconato et al, 2015).

## Proteomics sample preparation

Cells were treated with 1 µM VT107 or the equivalent % (v/v) DMSO for 1-h, 4-h, or 24-h then lysed by incubation in 4% sodium deoxycholate (Sigma-Aldrich) at 95 °C for 5 min. Lysates were tip-probe sonicated (2 × 7 s) at 4 °C and centrifuged (18,000 × g and 4 °C) for 10 min prior to protein quantification by BCA assay (ThermoFisher Scientific). Proteins in lysates (1 µg/ml) were reduced by combination with 10 mM tris-2-caboxyethyl phosphine (TCEP), then alkylated by incubation at 45 °C for 5 min in 40 mM 2-chloroacetamide (CAA). Reactions were cooled to room temperature then digested by incubation with sequencing-grade Trypsin (Sigma-Aldrich) and LysC (Wako, Japan) (1 protease:50 substrate proteins) at 37 °C for 16 h. Diluted peptides (50% in isopropanol) were acidified by combining with trifluoroacetic acid (TFA) to a final concentration of 1% (v/v), then loaded directly onto SDB-RPS (Sigma) micro-columns (packed in-house). Loaded columns were washed with isopropanol (supplemented with 1% TFA) followed by 5% acetonitrile (supplemented with 0.2% TFA) prior to elution of peptides in 80% acetonitrile (supplemented with 5% ammonium hydroxide). Eluted peptides were dried by vacuum centrifugation (45 °C for 45 min) and resuspended in 2% acetonitrile (supplemented with 0.1% TFA).

## Liquid chromatography tandem mass spectrometry

Peptides were analyzed on a Dionex 3500 nanoHPLC, coupled to an Orbitrap Exploris 480 mass spectrometer (ThermoFisher Scientific) by electrospray ionization in positive mode with 1.9 kV at 275 °C. Separation was achieved on a 40 × 75 µm column packed with C18AQ (1.9 µm; Dr Maisch, Germany) at 50 °C for 50 min at a flow rate of 300 nL/min. Peptides were eluted over a linear gradient of 3–40% of elution buffer (80% v/v acetonitrile, 0.1% v/v FA). Instruments were

operated in data-independent acquisition (DIA) mode with an MS1 spectrum acquired over the mass range 350–950 $m/z$ (60 K resolution, 1e6 automatic gain control (AGC) and 50 ms maximum injection time). This was followed by MS/MS analysis of 38 × 16 $m/z$ windows with a 1 $m/z$ overlap via HCD fragmentation (30 K resolution, 1e6 AGC) and automatic injection time. Raw MS data were processed using a Sepctronaut DirectDIA (version 14.8.201029.47784) with default parameters and searched against the human UniProt database (October 2020 release) and filtered to 1% FDR at the peptide spectral match and protein level. The data were searched with a maximum of 2 miss-cleavages, and methionine oxidation and protein N-terminus acetylation were set as variable modifications while carbamidomethylation of cysteine was set as a fixed modification. Quantification was performed using MS2-based extracted ion chromatograms employing 3–6 fragment ions >450 $m/z$ with automated fragment-ion interference removal as described previously (Bruderer et al, 2015).

## Genome-wide CRISPR/Cas9 screens

Two technical replicates were transduced with the genome-wide Brunello CRISPR/Cas9 library at 1000-fold representation and MOI of 0.3 (Doench et al, 2016). Positively transduced cells were selected with Puromycin (1 µg/ml) for 7 days and maintained at 1000-fold representation for the duration of the screen. Cells were treated with VT107 at its IC$_{50}$ concentration or DMSO (at the equivalent %v/v), for two weeks, then VT107 treatment-concentration increased to its cytostatic dose (and the equivalent %v/v DMSO) for 1–2 weeks. At the treatment endpoints, genomic DNA was extracted from cells at 1000-fold representation using the NucleoSpin Blood XL kit (Clonetech). Sequencing libraries were generated using one-step PCR to amplify the integrated sequence within the construct and the addition of a sample barcode and Illumina adapters as previously described (Tuano et al, 2023). PCR products were purified using AMPure beads (Beckman Coulter) and samples sequenced using HiSeq (Illumina). PoolQ (https://portals.broadinstitute.org/gpp/public/software/poolq) was used for deconvolution of FastQ files and alignment of sgRNA reads. The MAGeCK algorithm was used to identify enriched and depleted genes by comparing sgRNA distribution in drug-treated cells to DMSO control (Li et al, 2015).

## CRISPR/Cas9 genome editing

The lentiGuide-Puro (Addgene #52963) and lenti-Cas9-2A-Blast (Addgene #73310) vectors were from Addgene. Cells were virally transduced with lenti-Cas9-2A-Blast and selected with Blasticidin-S (3 µg/ml) for 7 days. For each gene-of-interest, two single-guide RNAs (sgRNAs) with the highest log$_2$fold-change values in the screen were selected from the Brunello CRISPR/Cas9 library. Guides were cloned into the lentiGuide-Puro vector as in (Sanjana et al, 2014). Cas9-expressing cells were virally transduced with the sgRNA-expression vectors and Puromycin (1 µg/ml) selection applied for 7 days. Evidence of CRISPR mutagenesis was obtained through immunoblotting or TIDE analysis of sanger sequencing data at sgRNA-targeted regions (Brinkman and van Steensel, 2019).

## In vitro drug efficacy studies

In all studies, cells were incubated at 37 °C and 5% CO$_2$, plating densities were empirically determined from the doubling rate of

individual cell lines and testing compounds applied as dose titrations in matrix format. For combination drug studies of VT107 (in NCI-H2052 or NCI-H226 cells), 500 cells were seeded per well in 384-well plates and allowed to adhere overnight. The next day, 5 doses of DMSO (within 0.05–0.2%v/v) or 7 doses of VT107 (within 2 nM–5 μM) combined with 9 different doses of trametinib (within 0.6–20 nM), AZD1480 (within 18.7 nM–4 nM), Bazedoxifene (within 500–5000 nM), BBI-608 (within 25–3500 nM), Ruxolitinib (within 1500–50,000 nM) or Tocilizumab (within 13.8–690 nM) were applied to the cells and allowed to incubate with the cells for 6 days. At the incubation endpoint, cells were fixed with 4% paraformaldehyde (Merck Life Sciences) and stained with 1 μg/ml DAPI (Sigma-Aldrich). Cell nuclei per condition were quantified using the CellInsight CX7 LED high-content analysis platform according to the manufacturer's protocol (Thermo Fisher Scientific). Dose-response matrices (%inhibition per treatment) were analyzed using Synergyfinder software (Ianevski et al, 2017). For combination treatments of VT108 with cobimetinib (assessed in 42 NSCLC cell lines) or trametinib (assessed in 10 mesothelioma lines), cells were seeded in 384-well plates (NSCLC cells at 250–1000 cells per well; mesothelioma cells at 500–1000 cells per well) and allowed to adhere overnight. The next day (Day 0), 9 doses of VT108 (within 0.05–10,000 nM) in combination with 9 doses of cobimetinib (within 0.4–3000 nM) or 5 doses of trametinib (within 0.5–10,000 nM) were added to the cells and allowed to incubate with the cells for a total of 5–7 days. Most of the cell lines were incubated with the compounds for 7 days, while a few cell lines that had faster doubling rates were incubated with the compounds for 5 or 6 days. At the end of the assay, cell proliferation was measured by CellTiter-Glo (CTG) Luminescent Cell Viability Assay Kit (Promega) according to the manufacturer's protocol. The percentage growth inhibition was calculated using Day 0 data as a negative control prior to generating dose-response matrices (% inhibition per treatment) for analysis using Synergyfinder.

### In vivo drug efficacy studies

All procedures related to animal handling, care, and treatment were performed according to the guidelines approved by the Institutional Animal Care and Use Committee (IACUC) of WuXi AppTec, following the guidance of the Association for Assessment and Accreditation of Laboratory Animal Care (AAALAC). Reference number for ethics approval: ON01-SH006-2023V1.0. Female BALB/c nude mice were supplied by Zhejiang Charles River Laboratory Animal Co., Ltd or Beijing Vital River Laboratory Animal Co., Ltd. Mice were aged between 6–9 weeks, weighing between 18 and 23 g, and were housed in polycarbonate ventilation cages at constant temperature (20–25 °C) and humidity (40–70%), with 3–4 animals per cage. Animals had free access to water and irradiation sterilized dry granule food, and corn cob bedding was changed twice per week. Trametinib was formulated in vehicle 1 (9% Castor oil+10% PEG-400 + 81% double distilled water), and VT108 was formulated in vehicle 2 (5% DMSO + 10% solutol +85% D5W; D5W = 5% glucose). The formulated compounds were orally administered once a day, every day. Animals in the combination groups received the compounds 30 min apart—trametinib first, followed by VT108. Animals in the vehicle group received vehicle 1 first and 30 min later vehicle 2. In the single

agent groups, animals either received formulated trametinib and then vehicle 2, 30 min apart or vehicle 1 and then formulated VT108. For the LU-01-0407 study, mice were implanted subcutaneously at the right flank with LU-01-0407 tumor slices (~30 mm³) for tumor development. Tumor-bearing animals were randomized, and treatment was started when the average tumor size reached 132 mm³. For the LU-01-0236 study, mice were implanted subcutaneously at the right flank with the tumor slices (20–30 mm³) for tumor development. Tumor-bearing animals were randomized, and treatment was started when the average tumor size reached 130 mm³. Tumor size and animal weights were monitored twice weekly. Tumor volume in mm³ was calculated using the formula: $V = 0.5\ a \times b^2$ where a and b are the long and short diameters of the tumor, respectively. Tumor growth inhibition (TGI) in percentage was calculated for each treatment group using the formula: $TGI\ (\%) = [1-(Ti-T0)/(Vi-V0)] \times 100$, where Ti is the average tumor volume of a treatment group on a given day, T0 is the average tumor volume of the treatment group on the day of treatment start, Vi is the average tumor volume of the vehicle control group on the same day as Ti, and V0 is the average tumor volume of the vehicle group on the day of treatment start. Statistical analysis of difference in the tumor volume among the groups were conducted on the data collected on the indicated treatment day (Day 39 for the LU-01-0407 study; Day 49 for the LU-01-0236 study).

### Study design and statistical analysis

Statistical methods were not used to determine sample size prior to experimentation and experiments were not performed in a blinded fashion. For in vivo experiments, tumor-bearing mice were randomized prior to treatment. A one-way ANOVA was performed to compare the tumor volume among groups, and when a significant F-statistics (a ratio of treatment variance to the error variance) was obtained, comparisons between groups were carried out with Games-Howell test. All data were analyzed using SPSS 17.0. $p < 0.05$ was considered statistically significant. Mass-spectrometry data was analyzed in Perseus and included median normalization and differential expression analysis using t-tests with multiple hypothesis correction using Benjamini-Hochberg FDR adjustment (Tyanova et al, 2016). To assess whether normally distributed data was statistically significantly different t-tests were perfromed using Graphpad Prism. To compare expression levels of different groups of genes in Fig. 4F, hypergeometric distribution analysis was performed. For combination therapy experiments, mean synergy scores were tested for statistical significance using the parametric bootstrapping method.

### Graphics

Figure 2A and the Synopsis image were created with BioRender.com.

## Data availability

The RNA-seq and ChIP-seq datasets produced in this study are available in NCBI GEO under the GEO accession number GSE269900. It is accessible via this weblink: https://www.ncbi.nlm.nih.gov/geo/query/acc.cgi?acc=GSE269900.

The source data of this paper are collected in the following database record: biostudies:S-SCDT-10_1038-S44319-024-00217-3.

## Peer review information

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

## Acknowledgements

We thank Avni Anand, Luis Malaver-Ortega, Henry Beetham, and members of the Harvey lab for discussions and comments on the manuscript, and A. Chand, T. John, and M. Shackleton for reagents. KFH was supported by a Senior Research Fellowship (APP1078220) and Investigator grant (APP1194467) from the National Health and Medical Research Council of Australia (NHMRC). AK was partly supported by an Australian Government Research Training Program Scholarship and Rosie Lew Peter MacCallum Cancer Foundation Postgraduate Award. This research was supported by a Lyall Watts Mesothelioma Research Grant from the Cancer Council Victoria (APP1157737) and the Peter MacCallum Cancer Foundation. We acknowledge the Monash Functional Genomics Platform and the following Peter MacCallum Cancer Centre core facilities: Flow Cytometry, Victorian Centre for Functional Genomics, Bioinformatics, Research Laboratory Support Services, and Centre for Advanced Histology and Microscopy, and support to them from the Peter MacCallum Cancer Foundation and the Australian Cancer Research Foundation.

## Author contributions

**Aishwarya Kulkarni**: Formal analysis; Investigation; Writing—original draft; Writing—review and editing. **Varshini Mohan**: Conceptualization; Investigation; Methodology; Writing—original draft. **Tracy T Tang**: Conceptualization; Formal analysis; Investigation; Writing—review and editing. **Leonard Post**: Conceptualization; Project administration; Writing—review and editing. **Yih-Chih Chan**: Data curation; Formal analysis; Writing—review and editing. **Murray Manning**: Investigation; Methodology; Writing—review and editing. **Niko Thio**: Formal analysis; Writing—review and editing. **Benjamin L Parker**: Formal analysis; Investigation; Writing—review and editing. **Mark A Dawson**: Conceptualization; Formal analysis; Writing—review and editing. **Joseph Rosenbluh**: Conceptualization; Investigation; Writing—review and editing. **Joseph HA Vissers**: Conceptualization; Supervision; Writing—review and editing. **Kieran F Harvey**: Conceptualization; Supervision; Funding acquisition; Writing—original draft; Project administration; Writing—review and editing.

Source data underlying figure panels in this paper may have individual authorship assigned. Where available, figure panel/source data authorship is listed in the following database record: biostudies:S-SCDT-10_1038-S44319-024-00217-3.

## Disclosure and competing interests statement

TT Tang and L Post report employment with Vivace Therapeutics and have equity interest in Vivace Therapeutics.

# Expanded View Figures

**Figure EV1.   The effect of TEAD inhibitors on the transcriptome and proteome of Hippo pathway mutant mesothelioma cells.**

(A) Charts of H2052 and H226 cell viability following treatment with different doses of VT107 for 4 days. $n = 3$ biological replicates. (B) Multi-dimensional scaling analysis plots of transcriptomes of biological replicates of RNA-seq analyses of H2052 and H226 cells treated with VT107 or DMSO. $n = 3$ biological replicates. (C) Gene set enrichment analysis plots of the Cordenonsi YAP signature in differentially expressed genes (VT107 vs DMSO) in H2052 cells and H226 cells. (D) Correlation plots comparing the transcriptomes of VT107-treated H2052 and H226 cells. The whole transcriptome is plotted in the left panel and significantly differentially expressed genes only in the right panel. Correlation was assessed using the Pearson's correlation coefficient. (E) A multi-dimensional scaling analysis plot of biological replicates of proteomes from H2052 cells treated with VT107 (TEADi) for 1, 4, or 24-h or with DMSO. $n = 5$ biological replicates. (F) Volcano plots of protein expression of H2052 cells following 1-h (1) or 4-h (ii) VT107 treatment, $n = 5$ biological replicates for each. Proteins whose abundance changed upon VT107 treatment are highlighted in red. (G) Immunoblots of lysates from the indicated cell lines, treated with 3 µM of VT107 or VT108 for 24 h and probed with the specified antibodies. Molecular mass markers are indicated. Data information: In (A), data are presented as mean ± SEM.

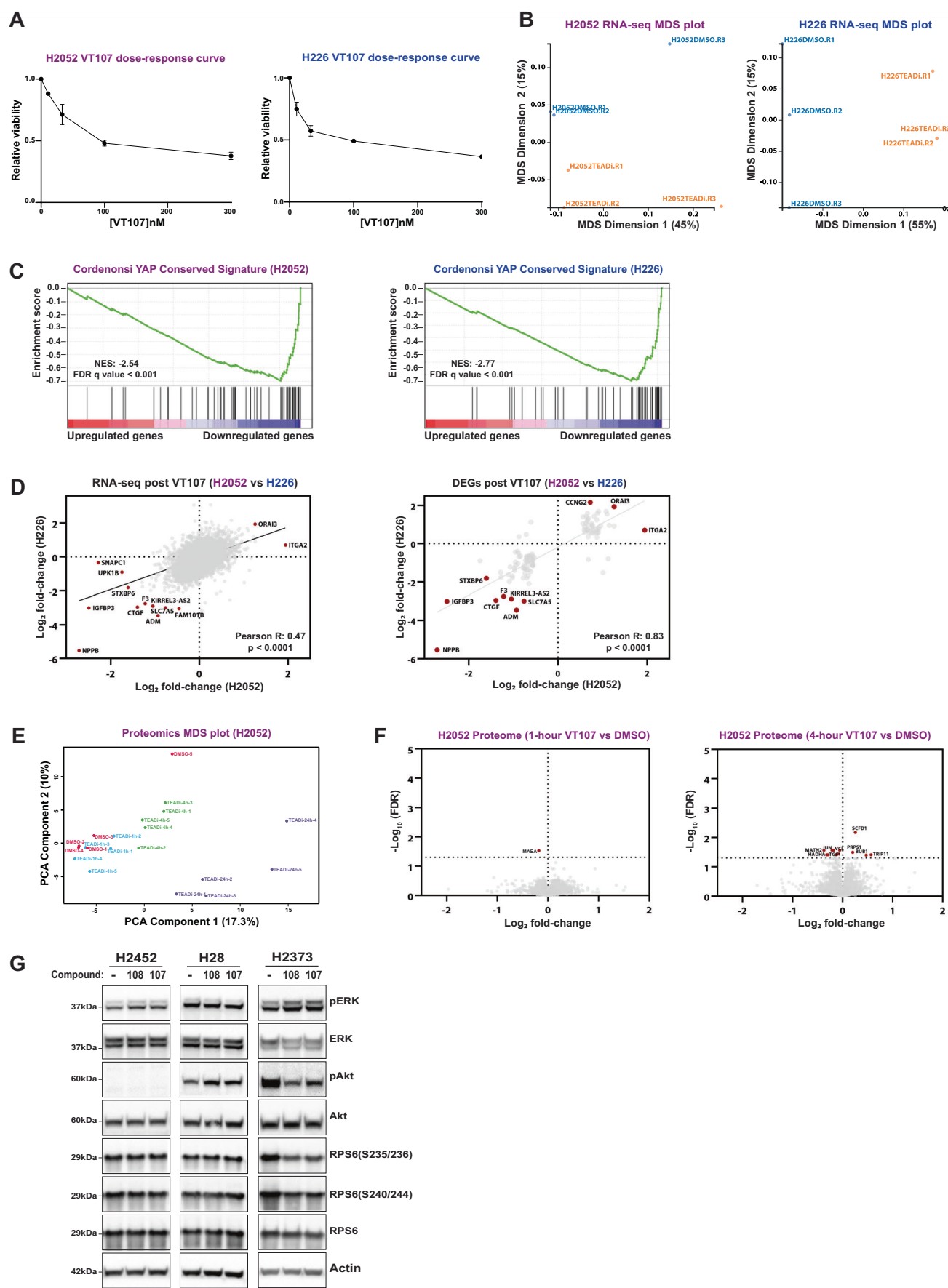

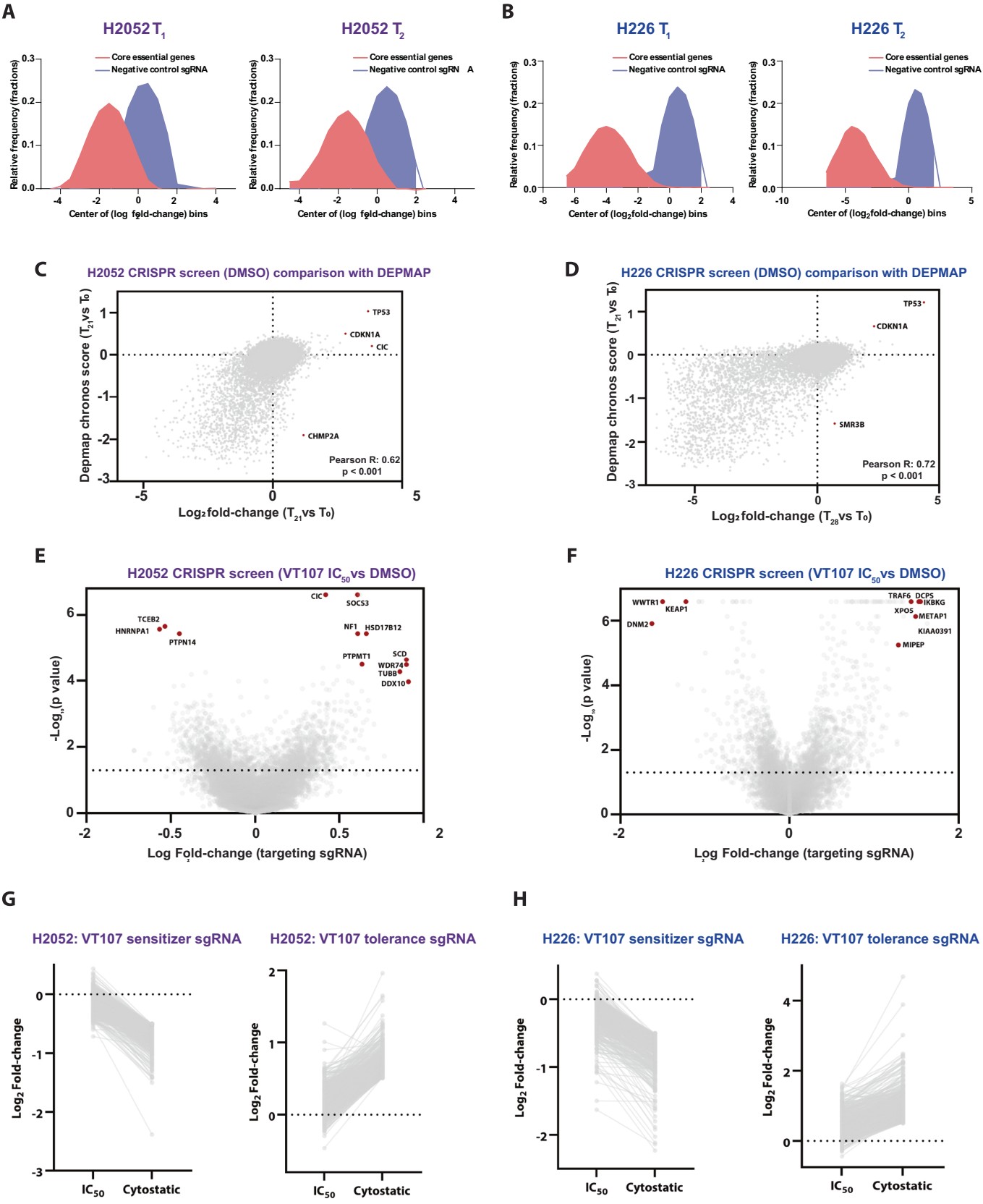

**Figure EV2. TEAD inhibitor CRISPR/Cas9 screens in mesothelioma cell lines.**

(A, B) Histograms representing the quantity of sgRNAs targeting core essential genes, or non-targeting control sgRNAs from the Brunello library, within the DMSO-treatment populations at the endpoint of $IC_{50}$ or cytostatic dose treatment with VT107 of H2052 cells (A) and H226 cells (B). (C, D) Correlation plots comparing data from our genome-wide CRISPR/Cas9 screens in DMSO-treated H2052 cells (C) or H226 cells (D) and data from the Depmap database. Correlation was assessed using the Pearson's correlation coefficient. (E, F) Volcano plots representing the overall change of targeting sgRNAs (per gene) in response to VT107 at the $IC_{50}$ dose during the CRISPR/Cas9 screens from H2052 cells ($IC_{50}$: 18 nM) (E) or H226 cells ($IC_{50}$: 33 nM) (F). (G, H) Comparison of the $\log_2$ fold-change of targeting gRNAs that conferred VT107 sensitivity or resistance at the $IC_{50}$ and cytostatic VT107 doses in H2052 cells (G) or H226 cells (H) during the CRISPR/Cas9 screens. Data information: In (E, F), significance was assessed using empirical Bayesian statistics.

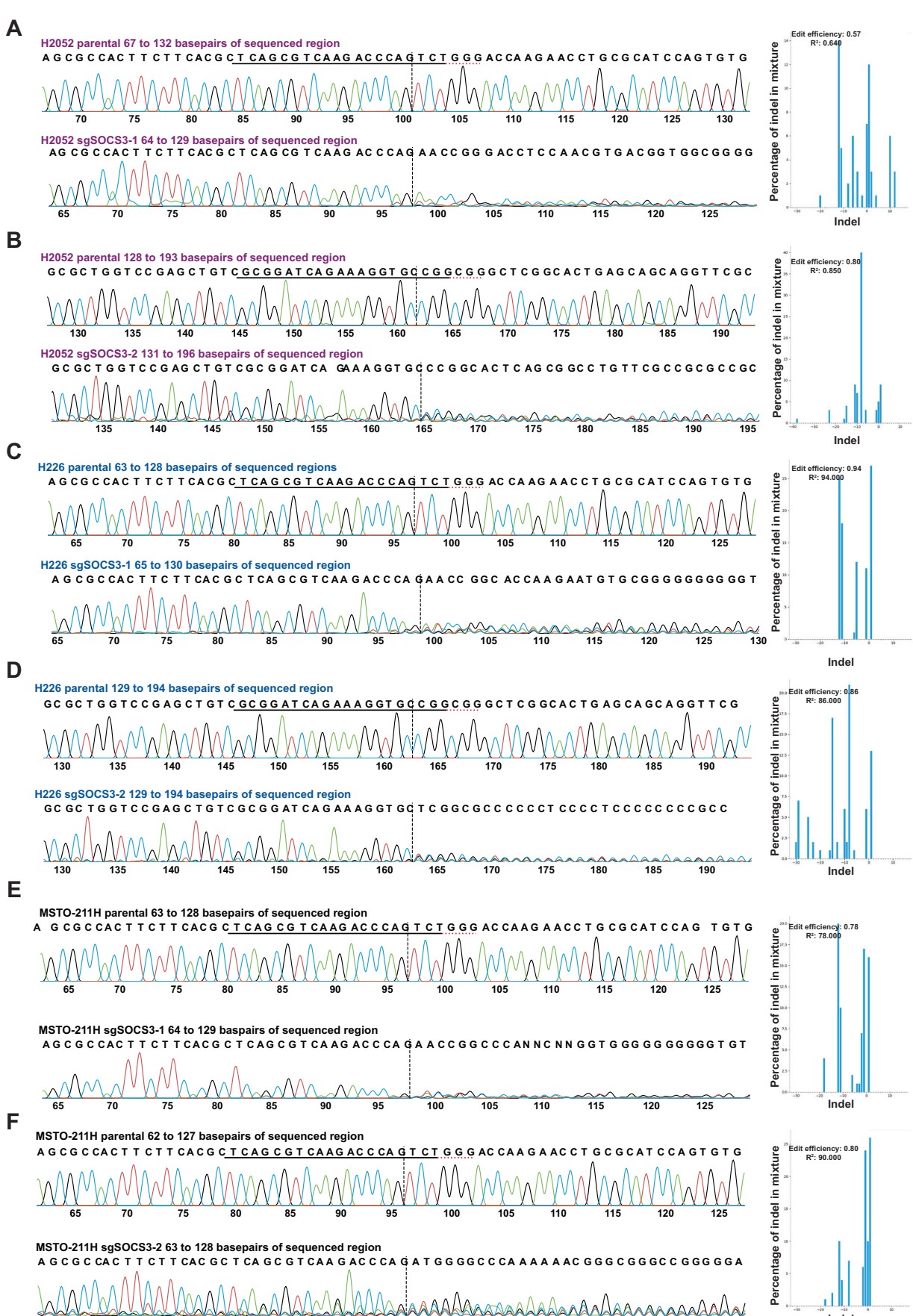

**Figure EV3. CRISPR/Cas9-induced mutagenesis of *SOCS3* in H2052, H226, and MSTO-211H cells.**

(A–F) DNA sequence alignment charts at independent sgRNA-targeted regions of the *SOCS3* gene in parental or sgRNA-expressing H2052 (**A**, **B**), H226 (**C**, **D**) and MSTO-211H (**E**, **F**) cells. Corresponding sgRNA and PAM sequences are underlined in black and red, respectively. TIDE-analysis calculations of CRISPR/Cas9-induced mutagenesis efficiency are indicated in bar charts on the right.

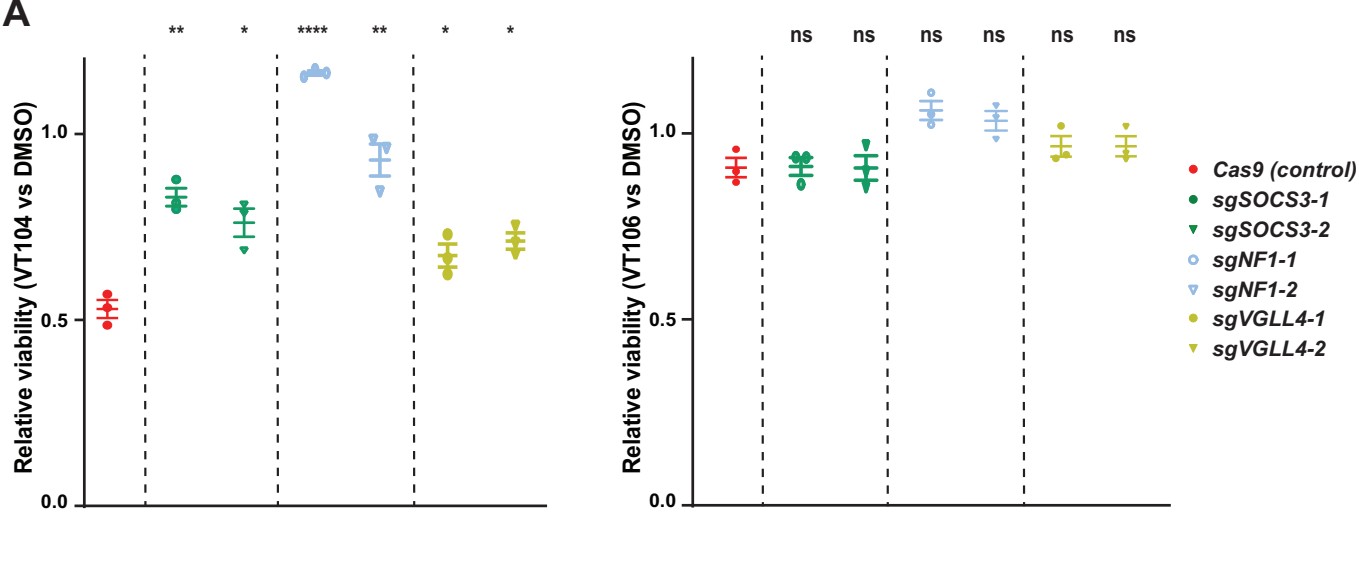

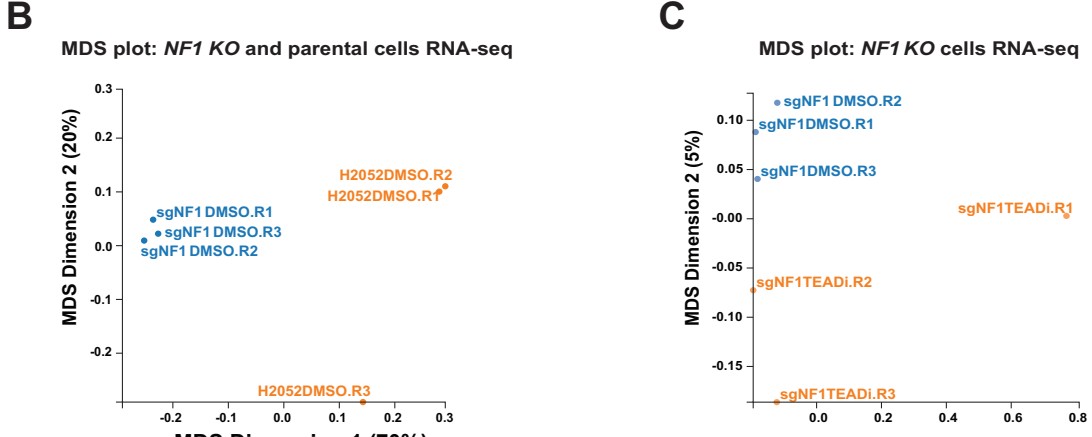

**Figure EV4. Mutation of *NF1*, *SOCS3*, or *VGLL4* in NCI-H2052 cells induces resistance to the VT104 TEAD inhibitor.**

(A) Charts of the impact of VT104 or VT106 on the viability of parental H2052 cells and *NF1*, *SOCS3*, or *VGLL4* mutant H2052 cells, as assessed by alamar blue assays. Cells were treated with 100 nM VT104 or VT106 for 4 days. $n = 3$ biological replicates. (B, C) Multi-dimensional scaling analysis plots comparing biological replicates of RNA-seq experiments performed on parental and *NF1* mutant H2052 cells treated with either DMSO or VT107 for 24 h. $n = 3$ biological replicates. Data information: In (A), data are presented as mean ± SEM. *$p < 0.05$, **$p < 0.01$, ****$p < 0.0001$, ns—not significant. (Student's t-tests). *P* values for VT104 chart were (from left to right): 0.00329; 0.03053; 0.00004; 0.00178; 0.03383; 0.01008. *P* values for VT106 chart were (from left to right): 0.89364; 0.95685; 0.0735; 0.15201; 0.48852; 0.48270.

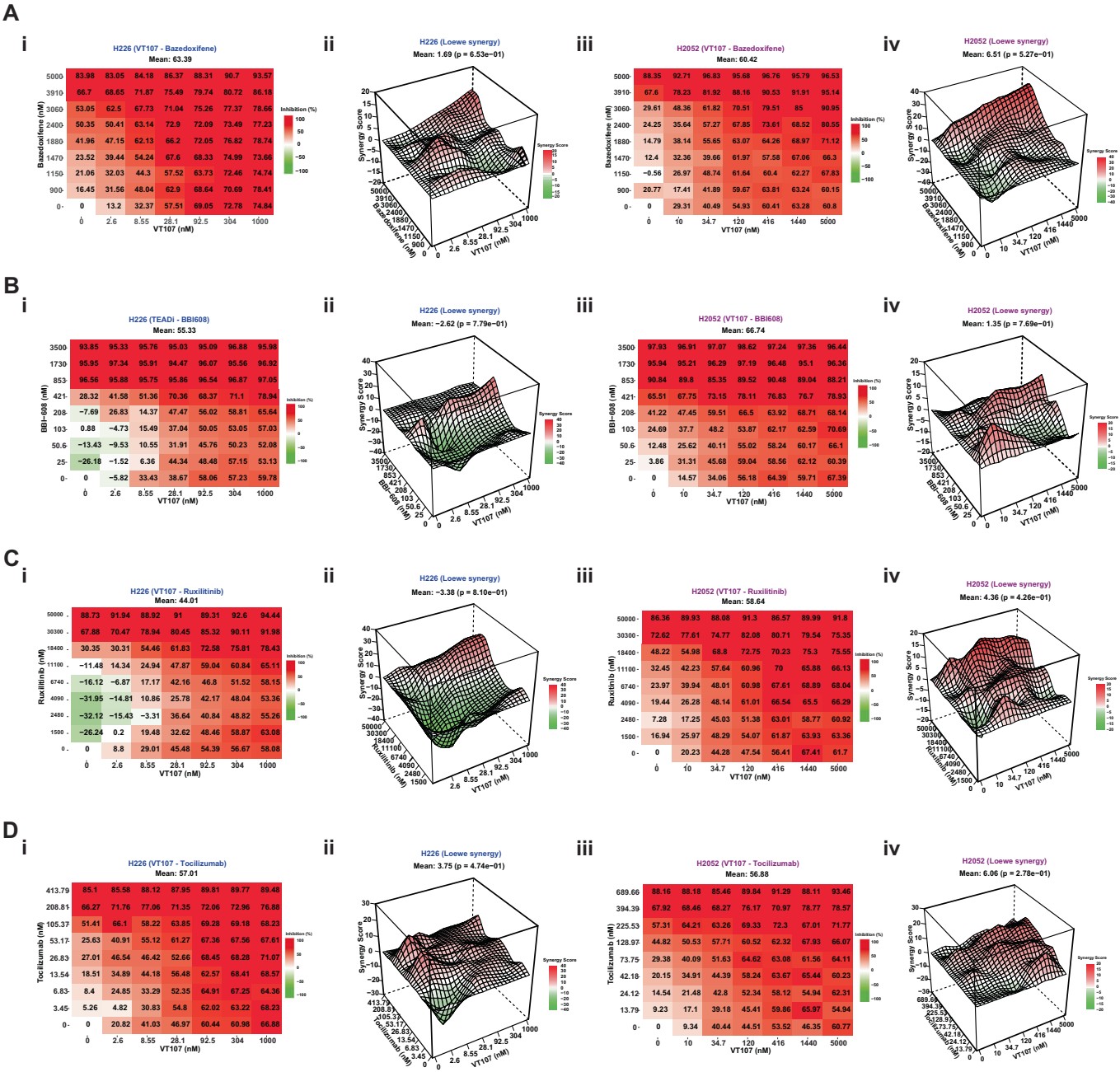

**Figure EV5. The impact of combined JAK/STAT pathway inhibitors and TEAD inhibitors on mesothelioma cells.**

(A–D) Dose-response matrices indicating the level of inhibition of cell number by treatment with different doses of VT107 and/or different JAK/STAT pathway inhibitors in H2052 cells (i) and H226 cells (iii). 3D synergy calculation topological maps indicating the type and degree of interaction between VT107 and different JAK/STAT pathway inhibitors in H2052 cell numbers (ii) and H226 cell numbers (iv). JAK/STAT pathway inhibitors that were used were Bazedoxifene (A), BBI-608 (B), Ruxilitinib (C) and Tocilizumab (D). In all assays, $n = 2$ biological replicates (2 technical replicates were performed for each biological replicate). Mean synergy scores were tested for statistical significance using the parametric bootstrapping method, $n = 2$ biological replicates.

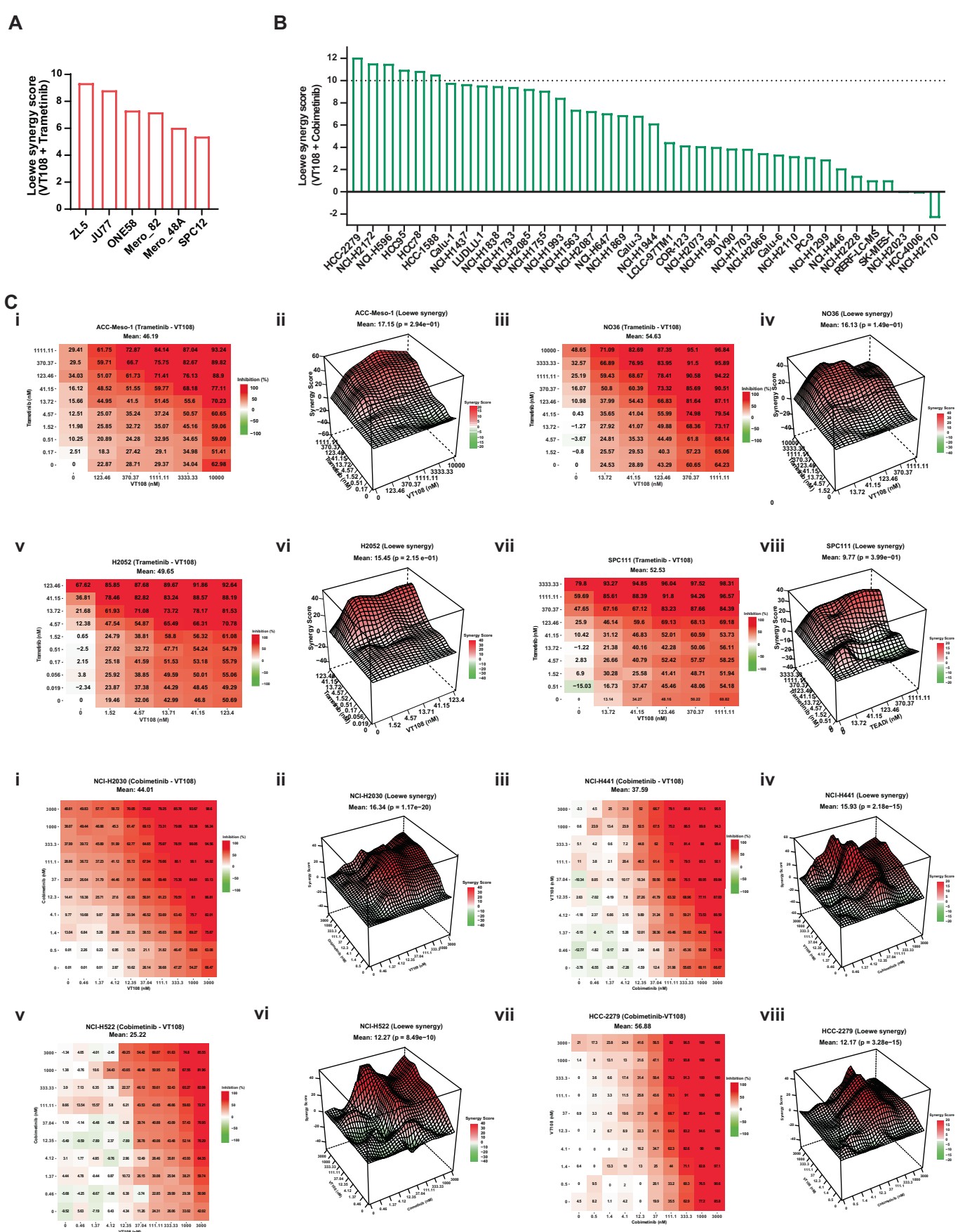

**Figure EV6.   The impact of combined MAPK pathway inhibitors and TEAD inhibitors on mesothelioma and NSCLC cells.**

(A, B) Bar charts indicating the interaction of combining VT108 and MEK inhibitors (trametinib or cobimetinib) on cell numbers of the indicated mesothelioma cell lines (A) or NSCLC cell lines (B). Synergistic interactions were indicated by synergy scores ≥10 and additive interactions by scores within the range of −10 to 10. $n = 2$ technical replicates. (C) Synergistic interactions between TEADi and MEKi in mesothelioma or NSCLC cell lines. Left panels: dose-response matrices indicating the level of inhibition of cell number by treatment of cells with the indicated doses of VT108 and either trametinib or cobimetinib. Right panels: Representative 3D synergy calculation topological maps indicating the type and degree of interaction between drugs on inhibition of cell number. Mean synergy scores were tested for statistical significance using the parametric bootstrapping method, $n = 2$ biological replicates.

