## [Peer Review File · EMBO Reports]

Identification of resistance mechanisms to small-molecule inhibition of TEAD-regulated transcription

Aishwarya Kulkarni, Varshini Mohan, Tracy Tang, Leonard Post, Yih-Chih Chan, Murray Manning, Niko Thio, Benjamin Parker, Mark Dawson, Joseph Rosenbluh, Joseph Vissers, and Kieran Harvey

Corresponding author(s): Kieran Harvey (kieran.harvey@petermac.org)

Review Timeline:

Submission Date:	27th Jun 24
Editorial Decision:	28th Jun 24
Revision Received:	11th Jul 24
Accepted:	17th Jul 24

Editor: Achim Breiling

Transaction Report: This manuscript was transferred to EMBO reports following peer review at The EMBO Journal.

Dear Dr. Harvey,

Thank you for transferring your revised manuscript to EMBO reports. I now went again through the manuscript, your point-by-point response and the referee reports from The EMBO Journal (attached below).

Based on these files and our previous e-mail conversation, I would like to invite you to provide a final revised manuscript as discussed, i.e. a more streamlined version (removing the ChIP-seq and RNA-seq from the Figures 5 and 6 and associated EV figures but keeping the MAPK pathway data in Figure 5 and the associated EV Figures) and a final p-b-p-response/rebuttal to the referee points.

Moreover, I have these editorial requests.

- Please provide the abstract written in present tense and with not more than 175 words.
- Please provide your final manuscript text file as one .docx formatted file including the legends for main figures and EV figures (but without the figures included).
- Please order the manuscript sections like this, using these names:
Title page - Abstract - Keywords - Introduction - Results - Discussion - Methods - Data availability section - Acknowledgements - Disclosure and Competing Interests Statement - References - Figure legends - Expanded View Figure legends
- Please remove the referee token from the Data availability section (DAS) and make sure the dataset is public latest upon online publication of the manuscript.
- Please upload individual production quality figure files as .eps, .tif, .jpg (one file per figure), of main figures and EV figures. Please upload these as separate, individual files upon re-submission.

The Expanded View format, which will be displayed in the main HTML of the paper in a collapsible format, has replaced the Supplementary information. You can submit up to 6 images as Expanded View. Please follow the nomenclature Figure EV1, Figure EV2 etc. The figure legend for these should be included in the main manuscript document file in a section called Expanded View Figure Legends after the main Figure Legends section. Additional Supplementary material should be supplied as a single pdf file labeled Appendix. The Appendix should have page numbers and needs to include a table of content on the first page (with page numbers) and legends for all content. Please follow the nomenclature Appendix Figure Sx, Appendix Table Sx etc. throughout the text, and also label the figures and tables according to this nomenclature.

- Please upload a completed author checklist (for EMBO reports as target journal), which you can download from our author guidelines (<https://www.embopress.org/page/journal/14693178/authorguide>). Please insert page numbers in the checklist to indicate where the requested information can be found in the manuscript.
- Regarding data quantification and statistics, please make sure that the number "n" for how many independent experiments were performed, their nature (biological versus technical replicates), the bars and error bars (e.g. SEM, SD) and the test used to calculate p-values is indicated in the respective figure legends (also for EV figures and all those in an Appendix). Please also check that all the p-values are explained in the legend, and that these fit to those shown in the figure. Please provide statistical testing where applicable. Please avoid the phrase 'independent experiment', but clearly state if these were biological or technical replicates. Please also indicate (e.g. with n.s.) if testing was performed, but the differences are not significant. In case n=2, please show the data as separate datapoints without error bars and statistics. See also: <http://www.embopress.org/page/journal/14693178/authorguide#statisticalanalysis>

If n<5, please show single datapoints for diagrams. Moreover:

- Please note that the panel 4G is mislabeled as panel 4E in the legend of Fig. 4. Please check.
- Please note that figure panels 7G-H are mentioned in the legend of Fig. 7, but the uploaded figure does not have such panels. Please check.
- Please note that Fig. EV1 has panels F and G, but the legend for Fig. EV1 does not mention panels F and G(i-ii). Please

check.

- Please provide the exact p values in the legends of Figs. 1h; 3g; 4f; 6d; 8a, c; EV 1e; EV 2c-d; EV 4.
- Please indicate the statistical test used for data analysis in the legends of Figs. 1d-e; 2b-c; 4a, c; 6a; EV 2e-f; EV 6d; EV 7a, c-e.
- Please note that information related to n is missing in the legends of Figs. 4f; 6d; EV 7f.
- Although 'n' is provided, please describe the nature of entity for 'n' in the legends of Figs. 1a-b; 3g; EV 1a(i-ii); EV 4.
- Please add to each legend (main, EV and Appendix figures, where applicable) a 'Data Information' section explaining the statistics used or providing information regarding replicates and scales. See:

- Please make sure that all figure panels are called out separately and sequentially. Presently, callouts for Fig. 4D seem missing. Moreover, there are callouts for panels 7G-H but no such panels. The EV figures should be called out as Figure EVx (not Fig. Sx). The Datasets should be called out as Dataset EVx (see below). Please check.
- Please upload the original excel files as datasets using the name Dataset EVx and use this name also for their callouts. Please provide legends for these on the first TAB of the excel file. Finally, please do not provide legends for these files in the final manuscript text file.
- Please make sure that all the funding information is also entered into the online submission system and is complete and similar to the one in the manuscript text file (in the Acknowledgements). Presently, these grants are only mentioned in the acknowledgements: Australian Government Research Training Program Scholarship, Rosie Lew Peter MacCallum Cancer Foundation Postgraduate Award; the Peter MacCallum Cancer Foundation.

- Please use our reference format:

- We would encourage you to use 'Structured Methods', our new Materials and Methods format. According to this format, the Materials and Methods section should include a Reagents and Tools Table (listing key reagents, experimental models, software, and relevant equipment and including their sources and relevant identifiers), uploaded as separate file, followed by a Methods and Protocols section in which we encourage the authors to describe their methods using a step-by-step protocol format with bullet points, to facilitate the adoption of the methodologies across labs. More information on how to adhere to this format as well as downloadable templates (.doc) for the Reagents and Tools Table can be found in our author guidelines (section 'Structured Methods'):

- We now request the publication of original source data with the aim of making primary data more accessible and transparent to the reader. Our source data coordinator will contact you to discuss which figure panels we would need source data for and will also provide you with helpful tips on how to upload and organize the files.

In addition, I would need from you:

- a short, two-sentence summary of the manuscript
- three to four short bullet points (two lines) that highlighting the key findings of your study
- a schematic summary figure (in jpeg or tiff format with the exact width of 550 pixels and a height of not more than 400 pixels) that can be used as a visual synopsis on our website.

I look forward to seeing a revised version of your manuscript when it is ready. Please let me know if you have questions or comments regarding the revision.

Kind regards,

Achim

Referee #2:

The authors have submitted a revised manuscript that is improved, modestly. However, it seems that they mostly responded to my comments by stating either they don't know the answer or did not want to do the experiment(s). This is a bit unusual in my experience and I am not able to make a definitive conclusion based on this approach taken by the authors. It seems my original questions and advice still stand.

Referee #3:

1. As the author described, the TEAD inhibitor consists of a palmitoylation inhibitor and a YAP-TEAD protein-protein interaction inhibitor. While this study focused on the palmitoylation inhibitor VT107, it is not clear what the distinctive phenotype of VT107 is compared to other palmitoylation inhibitors as well as YAP-TEAD protein-protein interaction inhibitors. It should be determined whether the results shown by the authors are class effects, compound-specific results, or a general phenomenon associated with TEAD inhibitors.
2. While TEAD palmitoylation inhibitors induced AKT activation (Sun et al., Reference #76), this was not observed in this study. What is the mechanism behind this difference?
3. The main finding of MAPK signal activation following VT107 treatment was not satisfactory supported by the results. In Figure 1E, the upregulation of KRAS, IL15, IL2, and LEF1 signaling does not necessarily indicate activation of the MAPK signal. Consistently, the phosphorylation status of the MAPK signal was unchanged in Figure 3. For example, was the readout protein DUSP6 downregulated following VT107 treatment? Additionally, how were the 44 MAPK genes selected?

Please find below responses (in blue) to Reviewer comments on the second round of review of our manuscript (EMBOJ-2023-115312). Please also find responses to Editor requests now that the manuscript has been transferred to EMBO Reports.

Referee #2:

The authors have submitted a revised manuscript that is improved, modestly. However, it seems that they mostly responded to my comments by stating either they don't know the answer or did not want to do the experiment(s). This is a bit unusual in my experience and I am not able to make a definitive conclusion based on this approach taken by the authors. It seems my original questions and advice still stand.

We are surprised and disappointed by the Reviewer's comments, given that we provided a point-by-point response to the EMBOJ Editor and Reviewers 2 and 3 before we invested almost 12 months in revising the manuscript according to this plan and their response to this proposed plan last year was: *"I think this would probably cover most of the main issues, but would need to see the final response and data to be 100% certain."*

Referee #3:

1. As the author described, the TEAD inhibitor consists of a palmitoylation inhibitor and a YAP-TEAD protein-protein interaction inhibitor. While this study focused on the palmitoylation inhibitor VT107, it is not clear what the distinctive phenotype of VT107 is compared to other palmitoylation inhibitors as well as YAP-TEAD protein-protein interaction inhibitors. It should be determined whether the results shown by the authors are class effects, compound-specific results, or a general phenomenon associated with TEAD inhibitors.

Following consultation with the EMBOJ Editor in response to this comment, we agreed to study two further TEAD inhibitor compounds. We included this new data in the revised manuscript for EMBOJ. In the current EMBO Reports manuscript this data is in EV4A.

2. While TEAD palmitoylation inhibitors induced AKT activation (Sun et al., Reference #76), this was not observed in this study. What is the mechanism behind this difference?

This point was not raised by the Reviewer in the initial review, nor when a point by point response and proposed revision plan was provided to the Reviewer and is beyond the scope of the current study.

3. The main finding of MAPK signal activation following VT107 treatment was not satisfactory supported by the results. In Figure 1E, the upregulation of KRAS, IL15, IL2, and LEF1 signaling does not necessarily indicate activation of the MAPK signal. Consistently, the phosphorylation status of the MAPK signal was unchanged in Figure 3. For example, was the readout protein DUSP6 downregulated following VT107 treatment? Additionally, how were the 44 MAPK genes selected?"

The comment from the Reviewer reflects a difference of opinion and indicates that they have not appreciated one of the major findings of our revised manuscript.

EMBO Reports Editor

Dear Dr. Harvey,

Thank you for transferring your revised manuscript to EMBO reports. I now went again through the manuscript, your point-by-point response and the referee reports from The EMBO Journal (attached below).

Based on these files and our previous e-mail conversation, I would like to invite you to provide a final revised manuscript as discussed, i.e. a more streamlined version (removing the ChIP-seq and RNA-

seq from the Figures 5 and 6 and associated EV figures but keeping the MAPK pathway data in Figure 5 and the associated EV Figures) and a final p-b-p-response/rebuttal to the referee points. We have made the requested changes to the manuscript. Please note, the MAPK pathway western blots have been moved to Figs. 1 and EV1.

Moreover, I have these editorial requests.

- Please provide the abstract written in present tense and with not more than 175 words. We have done this.

- Please provide your final manuscript text file as one .docx formatted file including the legends for main figures and EV figures (but without the figures included). We have done this.

- Please order the manuscript sections like this, using these names:
Title page - Abstract - Keywords - Introduction - Results - Discussion - Methods - Data availability section - Acknowledgements - Disclosure and Competing Interests Statement - References - Figure legends - Expanded View Figure legends
We have done this.

- Please remove the referee token from the Data availability section (DAS) and make sure the dataset is public latest upon online publication of the manuscript. We have done this and also removed the new ChIP-seq and RNA-seq that will not be published in the revised manuscript.

- Please upload individual production quality figure files as .eps, .tif, .jpg (one file per figure), of main figures and EV figures. Please upload these as separate, individual files upon re-submission. We have done this.

The Expanded View format, which will be displayed in the main HTML of the paper in a collapsible format, has replaced the Supplementary information. You can submit up to 6 images as Expanded View. Please follow the nomenclature Figure EV1, Figure EV2 etc. The figure legend for these should be included in the main manuscript document file in a section called Expanded View Figure Legends after the main Figure Legends section. Additional Supplementary material should be supplied as a single pdf file labeled Appendix. The Appendix should have page numbers and needs to include a table of content on the first page (with page numbers) and legends for all content. Please follow the nomenclature Appendix Figure Sx, Appendix Table Sx etc. throughout the text, and also label the figures and tables according to this nomenclature. We have done this.

See also the guidelines for figure legend

preparation: <https://www.embopress.org/page/journal/14693178/authorguide#figureformat>

- Please upload a completed author checklist (for EMBO reports as target journal), which you can download from our author guidelines (<https://www.embopress.org/page/journal/14693178/authorguide>). Please insert page numbers in the

checklist to indicate where the requested information can be found in the manuscript.

We have done this.

- Regarding data quantification and statistics, please make sure that the number "n" for how many independent experiments were performed, their nature (biological versus technical replicates), the bars and error bars (e.g. SEM, SD) and the test used to calculate p-values is indicated in the respective figure legends (also for EV figures and all those in an Appendix). Please also check that all the p-values are explained in the legend, and that these fit to those shown in the figure. Please provide statistical testing where applicable. Please avoid the phrase 'independent experiment', but clearly state if these were biological or technical replicates. Please also indicate (e.g. with n.s.) if testing was performed, but the differences are not significant. In case n=2, please show the data as separate datapoints without error bars and statistics. See also:

<http://www.embopress.org/page/journal/14693178/authorguide#statisticalanalysis>

We have done this.

If $n < 5$, please show single datapoints for diagrams. Moreover:

- Please note that the panel 4G is mislabeled as panel 4E in the legend of Fig. 4. Please check.

- Please note that figure panels 7G-H are mentioned in the legend of Fig. 7, but the uploaded figure does not have such panels. Please check.

- Please note that Fig. EV1 has panels F and G, but the legend for Fig. EV1 does not mention panels F and G(i-ii). Please check.

- Please provide the exact p values in the legends of Figs. 1h; 3g; 4f; 6d; 8a, c; EV 1e; EV 2c-d; EV 4.

- Please indicate the statistical test used for data analysis in the legends of Figs. 1d-e; 2b-c; 4a, c; 6a; EV 2e-f; EV 6d; EV 7a, c-e.

- Please note that information related to n is missing in the legends of Figs. 4f; 6d; EV 7f.

- Although 'n' is provided, please describe the nature of entity for 'n' in the legends of Figs. 1a-b; 3g; EV 1a(i-ii); EV 4.

We have gone through this carefully and made all necessary changes. Please note, as discussed with the Editor some figures were removed.

- Please add to each legend (main, EV and Appendix figures, where applicable) a 'Data Information' section explaining the statistics used or providing information regarding replicates and scales. See:

We have done this.

- Please make sure that all figure panels are called out separately and sequentially. Presently, callouts for Fig. 4D seem missing. Moreover, there are callouts for panels 7G-H but no such panels. The EV figures should be called out as Figure EVx (not Fig. Sx). The Datasets should be called out as Dataset EVx (see below). Please check.

We have done this.

- Please upload the original excel files as datasets using the name Dataset EVx and use this name also for their callouts. Please provide legends for these on the first TAB of the excel file. Finally, please do not provide legends for these files in the final manuscript text file.

We have done this.

- Please make sure that all the funding information is also entered into the online submission system and is complete and similar to the one in the manuscript text file (in the Acknowledgements). Presently, these grants are only mentioned in the acknowledgements: Australian Government Research Training Program Scholarship, Rosie Lew Peter MacCallum Cancer Foundation Postgraduate Award; the Peter MacCallum Cancer Foundation.

Please note, the awards mentioned above (Australian Government Research Training Program Scholarship, Rosie Lew Peter MacCallum Cancer Foundation Postgraduate Award; the Peter

MacCallum Cancer Foundation) do not have associated funding numbers and so cannot be provided. We have provided all grant funding numbers where they are available.

- Please use our reference format:

We have done this.

- We would encourage you to use 'Structured Methods', our new Materials and Methods format. According to this format, the Materials and Methods section should include a Reagents and Tools Table (listing key reagents, experimental models, software, and relevant equipment and including their sources and relevant identifiers), uploaded as separate file, followed by a Methods and Protocols section in which we encourage the authors to describe their methods using a step-by-step protocol format with bullet points, to facilitate the adoption of the methodologies across labs. More information on how to adhere to this format as well as downloadable templates (.doc) for the Reagents and Tools Table can be found in our author guidelines (section 'Structured Methods'):

As discussed with the Editor, we have not done this.

- We now request the publication of original source data with the aim of making primary data more accessible and transparent to the reader. Our source data coordinator will contact you to discuss which figure panels we would need source data for and will also provide you with helpful tips on how to upload and organize the files.

We have done this.

In addition, I would need from you:

- a short, two-sentence summary of the manuscript
- three to four short bullet points (two lines) that highlighting the key findings of your study
- a schematic summary figure (in jpeg or tiff format with the exact width of 550 pixels and a height of not more than 400 pixels) that can be used as a visual synopsis on our website.

We have done this.

Kieran Harvey
Peter MacCallum Cancer Centre
Organogenesis and Cancer
305 Grattan St
Melbourne, Victoria 3000
Australia

Dear Dr. Harvey,

I am very pleased to accept your manuscript for publication in the next available issue of EMBO reports. Thank you for your contribution to our journal.

Yours sincerely,
